# MindAttention: Foveated Visual Encoding for Neural Response Synthesis and Concept-selective Region Localization

## Abstract

Generative models for synthesizing brain activity have emerged as powerful tools for mapping cortical functions. However, current approaches largely neglect the foveated nature of human vision, often assuming uniform processing across the entire visual field. This assumption fails when visual stimuli are complex, as the human brain prioritizes attended regions while suppressing peripheral distractions. Consequently, existing methods suffer from biased neural predictions and inaccurate localization. To bridge this gap, we propose *MindAttention*, a brain visual encoding framework that explicitly models the interaction between foveal fixations and neural encoding. Unlike standard global processing models, our approach incorporates a gaze-conditioned mechanism: it dynamically emphasizes visual features falling within the foveal field to drive simulated cortical responses, mimicking the biological constraint that high-level semantic processing is fovea-dependent. Experiments demonstrate that *MindAttention* achieves superior performance in localization accuracy compared to existing baselines. Instead of claiming full mechanistic interpretability, we suggest that incorporating these spatial attention constraints offers enhanced biological plausibility, establishing a more reliable paradigm for the data-driven exploration of brain concept maps.

## 1 Introduction

Human visual perception is an extremely complex system. Extensive research (Grill-Spector & Weiner, 2014; Kanwisher et al., 1997; Downing et al., 2001; Epstein & Kanwisher, 1998) has shown that the brain cortex exhibits concept selectivity in processing visual stimulus inputs. Specifically, when receiving stimuli belonging to a particular concept, specific brain regions are significantly activated. In tradition, localizing concept-selective regions relies on the *data collection with statistical analysis*. The experimental paradigm requires substantial investment in time, equipment, and resources, leading to lengthy research cycles and limited exploration of open-world concept categories. Inspired by the application of artificial intelligence for science (AI4S) research (Senior et al., 2020; Jumper et al., 2021), deep learning-based brain visual encoding models hold promise as a novel and efficient paradigm for data-driven concept region localization (Bao et al., 2025a).

Functional magnetic resonance imaging (fMRI) is favored for brain visual encoding models due to its non-invasive nature and high spatial resolution, effectively capturing neural responses to visual stimuli (Gu et al., 2022; Luo et al., 2023; Beliy et al., 2024; Xue et al., 2024; Bao et al., 2025a; Luo et al., 2025; Yu et al., 2025). These encoding models can be broadly classified into two paradigms: feature mapping and representation alignment. Feature mapping methods leverage powerful pre-trained vision models to extract hierarchical features from images, which are then regressed onto fMRI activity patterns using linear or nonlinear models. In contrast, representation alignment strategies often employ autoencoder architectures to learn a joint latent space for image-fMRI pairs. These models typically incorporate contrastive learning objectives to enforce cross-modal consistency, pulling corresponding representations of matching pairs closer while distancing those of non-matching pairs, thereby fostering more discriminative neural codes.

Despite their progress, these methods are predicated on a critical, biologically implausible assumption: that all regions of a visual stimulus contribute uniformly to the encoding of neural responses.

This premise starkly contrasts with fundamental principles of the human visual system, which is characterized by the high-acuity foveal region and the selective allocation of attention to task-relevant information(Rosenholtz, 2016). Decades of eye-tracking research have confirmed that human gaze is not uniform but follows consistent, preferential patterns(Larson & Loschky, 2009; Henderson, 2003). Consequently, by processing images in their entirety, existing models incorporate substantial irrelevant visual information, which not only degrades predictive accuracy but also obscures the neuro-computational mechanisms underlying perception. As illustrated in Figure 1, a participant's gaze may be directed towards other elements in the scene rather than the person. In this case, brain encoding models that rely on global image features erroneously attribute neural responses elicited by the attended object to the semantic category of the unattended person. This fundamental *synthesis-attention misalignment* between the nominal image content and the subject's perceptual experience introduces substantial bias, which not only degrades predictive accuracy but also obscures the neuro-computational mechanisms underlying perception.

To address this gap, we introduce **MindAttention**, a novel encoding framework that uniquely integrates the foveal attention mechanism into the image-to–fMRI mapping process. We hypothesize that high-level semantic information in the visual cortex is predominantly driven by stimuli within the foveal field of view. By prioritizing these attentionally salient regions, *MindAttention* enhances not only the biological plausibility but also the predictive performance of neural encoding models.

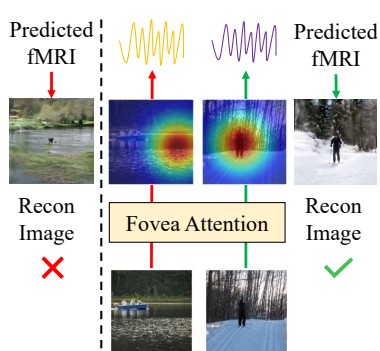

Figure 1: Example of visual spatial selectivity in the functional localization experiment. **Left**: Invalid stimuli. The background is more visually salient than the target. **Right**: Valid stimuli. The person is presented as the clear and unambiguous focus.

Our evaluations demonstrate that *MindAttention* consistently and significantly outperforms all baselines, achieving up to 3.6% relative improvement in voxel-wise correlation and 5.4% higher semantic alignment scores on average across subjects and visual areas. Notably, the performance gap widens in complex, multi-object scenes where attentional competition is high, confirming that our fovea-anchored mechanism effectively resolves the synthesis-attention misalignment inherent in global-feature models. These results further suggest that neural encoding models hold promise for achieving better data-driven localization of concept-selective brain regions.

In summary, our primary contributions are as follows:

- **Theoretically**, we reframe the role of attention in neural encoding by treating human gaze fixations as dynamic anchors that guide the formation of neural representations. This "attention-as-anchor" perspective provides a computational bridge for the biological mechanism of foveal vision, offering a new way to model how attention shapes neural activity.

- **Methodologically**, we propose *MindAttention*, a novel gaze-conditioned generative brain visual encoding framework that integrates fovea-guided feature selection and spatial transformation. By aligning visual inputs with human fixation patterns, *MindAttention* enables an interpretable and biologically grounded encoding of neural responses.

- **Empirically**, we demonstrate that *MindAttention* significantly surpasses state-of-the-art baselines across multiple visual areas in terms of both voxel-level prediction accuracy and semantic-level fidelity. These results validate the effectiveness and generalizability of incorporating foveal attention mechanisms into neural encoding models.

## 2 RELATED WORKS

**fMRI Visual Encoding Models.** Current fMRI visual encoding research largely follows two paradigms: discriminative modeling (Kay et al., 2008; Han et al., 2019; Gu et al., 2022; Luo et al., 2023; Beliy et al., 2024; Xue et al., 2024; Luo et al., 2025; Yu et al., 2025) and generative modeling (Bao et al., 2025a). The former maps visual stimulus representations to voxel-wise brain responses using regression models. Notably, seminal works have introduced biologically inspired

constraints to this mapping. Klindt et al. (2017) proposed factorizing model parameters to explicitly separate spatial location ("where") from feature tuning ("what"), while Lurz et al. (2020) further advanced this by employing Gaussian readout mechanisms to model the retinotopic properties of receptive fields. The latter seeks to synthesize visual content conditional on fMRI signals, leveraging generative modeling such as diffusion models (Ho et al., 2020; Ramesh et al., 2022; Peebles & Xie, 2023). In contrast to prior work, we advance this spatial-feature decoupling from the readout layer to the core of representation learning. Unlike these approaches which typically fit receptive fields to fixed features, we embed a dynamic, content-driven Gaussian attention mechanism directly into the encoder backbone to simulate active visual sampling. We present a high-fidelity generative encoding framework that substantially enhances reconstruction fidelity and semantic alignment through the integration of multi-scale brain region features, cross-modal alignment cues, and diffusion priors.

**fMRI Visual Semantic Decoding.** Recent advances in brain decoding have captured high-level semantic content, enabling the reconstruction of perceived visual scenes from fMRI activity, such as images (Takagi & Nishimoto, 2023; Chen et al., 2023a; Lu et al., 2023; Ozcelik & VanRullen, 2023; Mai & Zhang, 2023; Liu et al., 2023; Scotti et al., 2023; Gong et al., 2024b; Scotti et al., 2024; Gong et al., 2025; Bao et al., 2025b) and videos (Chen et al., 2023b; Lu et al., 2024; Gong et al., 2024a; Fosco et al., 2024; Yeung et al., 2025). In our work, we repurpose such pre-trained decoding models as components of our semantic-level evaluation pipeline, leveraging their powerful visualization capability to assess the fidelity of synthetic fMRI with respect to the original one.

**Preliminary on Human Visual Foveation Mechanism.** The human visual system is fundamentally foveated: high-acuity vision is restricted to the central $1°$–$2°$ of the retina, the fovea, where photoreceptor density peaks and cortical magnification is highest (Wässle et al., 1991; Daniel & Whitteridge, 1993). Outside this region, spatial resolution drops sharply, and semantic recognition becomes increasingly dependent on saccadic eye movements to bring targets of interest into foveal view (Larson & Loschky, 2009). Critically, neuroimaging and eye-tracking studies confirm that neural responses in high-level visual areas (e.g., FFA, PPA, EBA) are strongly modulated by foveal input – only stimuli fixated within the central foveal field reliably elicit robust, category-selective activation (Kay et al., 2015; Allen et al., 2021). This physiological constraint implies that conventional encoding models, which rely on full-image features, misrepresent the brain's true input sampling strategy. In particular, current encoding models introduce bias from non-foveal regions that are either neurally suppressed or only weakly encoded (Rolfs et al., 2011). Building on this principle, we treat the foveal fixation point not merely as a behavioral artifact but as a conceptual anchor of attention, which gates semantic-level cortical representations. We explicitly embed this mechanism into *MindAttention* to better align synthetic visual inputs with biological encoding priors.

## 3 METHODS

The proposed *MindAttention* reconstructs individualized cortical responses by aligning visual encoding with human foveal attention. Below, we first motivate this fovea-aligned design from neurobiological principles and provide an overview of the framework. We next elaborate on each core component, namely the Fovea-Guided Encoder, the fMRI Variational Autoencoder, and the Diffusion-Based Generator, and explain how they work together to enable attention-grounded, subject-specific response synthesis. Finally, we demonstrate that calibrated image conditions can serve as personalized priors to further synthesize stimulus-evoked fMRI signals.

### 3.1 MOTIVATION AND OVERVIEW

Reconstructing individualized brain cortical responses to visual stimuli is not merely a technical mapping task but rather a problem of biological alignment. Although recent advances in brain encoding models have achieved impressive predictive accuracy, most approaches rely on the assumption that the entire visual field contributes uniformly to high-level neural representations. This assumption overlooks a fundamental property of human vision, which is intrinsically fovea-centric (Rosenholtz, 2016; Larson & Loschky, 2009; Henderson, 2003). Evidence from both neurophysiology and psychophysics demonstrates that semantic-level encoding in ventral visual regions, including V4, LOC, and IT, is driven predominantly by information within the central 2–5 degrees of visual angle(Grill-Spector & Malach, 2004; Larson & Loschky, 2009). Peripheral inputs, even when visually salient, are largely suppressed or represented only at coarse and non-semantic levels(Larson & Loschky, 2009; Hasson et al., 2002). Models that disregard this spatial gating mechanism and

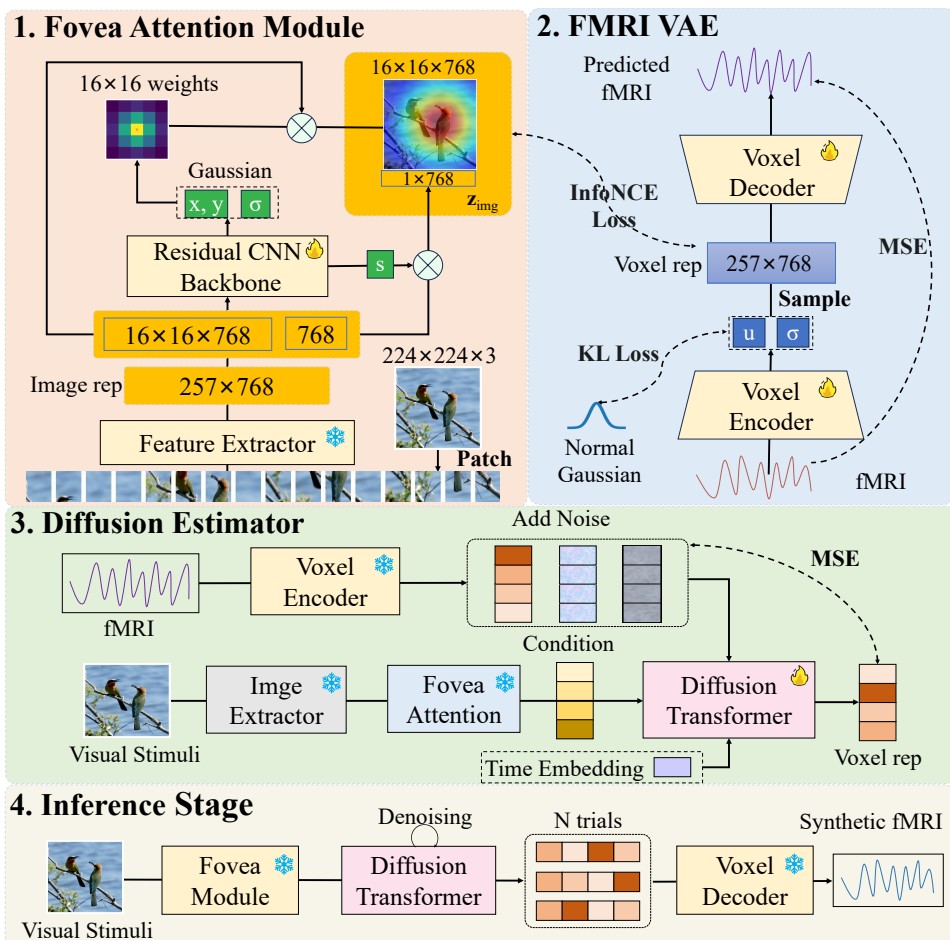

Figure 2: Overview of the *MindAttention* framework. This schematic shows the end-to-end framework for synthesizing individualized, attention-grounded cortical responses. A fovea-guided encoder extracts visual embeddings, which are aligned with fMRI representations through contrastive learning. A diffusion model then leverages these aligned features to generate high-fidelity, subject-specific fMRI maps for novel images.

incorporate entire-image features into regressors or generative encoders encounter what we refer to as the *synthesis–attention misalignment* problem. In this setting, the model is required to explain neural activity using visual signals that the brain itself does not utilize. Such a mismatch not only introduces bias and diminishes predictive fidelity but also compromises interpretability, as it becomes unclear whether the model captures genuine neural encoding processes or merely increases statistical correlations by exploiting information present in the image but irrelevant to human attention.

*MindAttention* is designed to resolve this mismatch at its source. It explicitly aligns visual feature extraction with human fixation behavior, ensuring spatial consistency between input representation and the actual encoding mechanism of the brain. As illustrated in Figure 2, the framework consists of three core components: **1) Fovea-Guided Visual Encoder** that dynamically focuses on local image regions based on predicted fixations, extracting visual features the brain *actually cares about*; **2) fMRI Variational Autoencoder** that learns a compact, structure-preserving latent space of neural responses; and **3) Diffusion-Based Conditional Generator** that synthesizes diverse, biologically plausible, individualized response patterns – built upon attention-aligned representations.

## 3.2 FOVEA MODULE FOR NEURALLY-ALIGNED VISUAL REPRESENTATION

To emulate the biological principles of human vision, we introduce the **Fovea Module**, a neural component designed to generate visual representations that mirror the mechanics of foveal perception. This module dynamically reweights image patch features to simulate cortical magnifica-

tion—producing a perceptual profile characterized by high acuity at a predicted attentional center and gradually diminishing resolution toward the periphery.

Given each data pair $(\mathbf{x}, \mathbf{y})$ from the subject-individual fMRI dataset $\mathcal{S}$, $\mathbf{x} \in \mathbb{R}^D$ (where D is typically tens of thousands of voxels) denotes preprocessed fMRI blood oxygenation level-dependent (BOLD) voxels and $\mathbf{y}$ denotes the corresponding visual stimuli, i.e., an image. Each input image is first processed by a vision backbone (e.g., ViT) to yield image representation $\mathbf{Y}_{\text{img}} \in \mathbb{R}^{(N+1) \times d}$, comprising $N$ image patch embeddings and a single [CLS] token that captures global semantics.

The Foveal Attention Module (FAM) operates as follows. First, the $N$ patch embeddings, i.e., $\mathbf{Y}_{\text{img}}^{1:N}$, are spatially rearranged into a 2D grid corresponding to their original image locations. A lightweight sub-network—formally defined as the Foveal Predictor (implemented via a CNN backbone)—takes the grid as input and regresses three key parameters that define the attentional field:

$$(\mu_x, \mu_y),\ \tilde{\ell},\ w_{\text{logit}} = \text{FovealPredictor}(\text{Grid}(\mathbf{Y}_{\text{img}}^{1:N})) \tag{1}$$

From these raw outputs, the interpretable parameters are derived as:

- **Attentional Center** $(\mu_x, \mu_y) \in [-1, 1]^2$: Normalized coordinates of the predicted fixation point representing the foveal centroid.

- **Spread Parameter** $\sigma = \exp(\tilde{\ell}) > 0$: Controls the spatial extent of the foveal region. Smaller $\sigma$ yields highly localized focus, whereas larger $\sigma$ produces broader attention.

- **Global Context Weight** $w_{\text{cls}} = \sigma(w_{\text{logit}}) \in [0, 1]$: A sigmoid-scaled weight modulating the contribution of the [CLS] token. Biologically, this mimics the "top-down" modulation mechanism, where holistic semantic information from higher visual areas (represented by the [CLS] token) helps constrain and guide local feature processing.

Each patch $i$ with normalized spatial coordinate $\mathbf{p}_i \in [-1, 1]^2$ is weighted via a 2D isotropic Gaussian kernel centered at $\boldsymbol{\mu} = (\mu_x, \mu_y)$. All weights are normalized across patches using Softmax:

$$w_i = \frac{\exp\left(-\frac{\|\mathbf{p}_i - \boldsymbol{\mu}\|^2}{2\sigma^2}\right)}{\sum_{j=1}^N \exp\left(-\frac{\|\mathbf{p}_j - \boldsymbol{\mu}\|^2}{2\sigma^2}\right)}. \tag{2}$$

The final fovea-modulated image representation $\mathbf{z}_{\text{img}} \in \mathbb{R}^d$ integrates localized high-acuity features with global context:

$$\mathbf{z}_{\text{img}} = \sum_{i=1}^N w_i \cdot \text{Patch}_i + w_{\text{cls}} \cdot \text{CLS}. \tag{3}$$

This integration strategy is critical for robust encoding. While the Gaussian-weighted patches capture fine-grained spatial details (simulating the high-resolution input of the fovea), the weighted [CLS] token preserves the global semantic gist. This dual-pathway approach computationally models the interaction between bottom-up sensory inputs and top-down contextual priors, ensuring that the final representation remains semantically coherent even when the foveal window is narrow.

To ensure biological plausibility and alignment with actual human visual processing, we train the entire system—including the backbone and Fovea Module—using a contrastive learning objective grounded in fMRI data. Specifically, we minimize the InfoNCE loss(Oord et al., 2018) between the foveated image embedding $\mathbf{z}_{\text{img}}$ and its corresponding fMRI-derived neural embedding $\mathbf{X}_{\text{fMRI}}$ of $\mathbf{x}$:

$$\mathcal{L}_{\text{InfoNCE}} = -\mathbb{E}_{(\mathbf{x}, \mathbf{y}) \sim \mathcal{S}} \left[ \log \frac{\exp(\text{sim}(\mathbf{X}_{\text{fMRI}}, \mathbf{z}_{\text{img}})/\tau)}{\sum_{\mathbf{z}'_{\text{img}}} \exp(\text{sim}(\mathbf{X}_{\text{fMRI}}, \mathbf{z}'_{\text{img}})/\tau)} \right], \tag{4}$$

where $\text{sim}(\cdot, \cdot)$ denotes cosine similarity, the denominator sums over all image embeddings in the batch (one positive, rest negative), and $\tau$ is a temperature hyperparameter.

This framework ensures that the learned visual representations not only mimic the spatial selectivity of human foveal vision but are also neurally aligned with real brain activity patterns.

## 3.3 FMRI VARIATIONAL AUTOENCODER

Despite the use of attention-aligned visual features, directly modeling high-dimensional fMRI voxel responses remains inherently noisy, computationally inefficient, and susceptible to overfitting — particularly given the limited sample sizes typical in neuroimaging studies. More critically, such direct modeling fails to capture or respect the brain's intrinsic functional organization. To overcome these limitations, we propose a dedicated fMRI Variational Autoencoder (fMRI-VAE) architecture composed of a paired encoder-decoder framework that explicitly learns a compressed, neurobiologically meaningful latent representation of brain activity.

The encoder, denoted as $q_\phi(\mathbf{X}_{\text{fMRI}}|\mathbf{x})$, maps the high-dimensional fMRI voxel vector $\mathbf{x}$ into a low-dimensional latent space $\mathbf{X}_{\text{fMRI}} \in \mathbb{R}^{(N+1)d}$. The encoder is implemented as a feedforward neural network with parameters $\phi$, outputting the mean $\mu_\phi(\mathbf{x})$ and log-variance $\log \sigma_\phi^2(\mathbf{x})$ of a diagonal Gaussian distribution from which the latent code $\mathbf{z}$ is sampled via the reparameterization trick:

$$\mathbf{X}_{\text{fMRI}} = \mu_\phi(\mathbf{x}) + \sigma_\phi(\mathbf{x}) \cdot \epsilon, \quad \epsilon \sim \mathcal{N}(\mathbf{0}, \mathbf{I})$$

The decoder, $p_\theta(\hat{\mathbf{x}}|\mathbf{X}_{\text{fMRI}})$, reconstructs the original fMRI response from the latent code $\mathbf{z}$ using another multi-layer perceptron with parameters $\theta$, yielding a reconstructed voxel vector $\hat{\mathbf{x}} \in \mathbb{R}^D$.

The model is trained end-to-end by optimizing the following variational lower bound (ELBO):

$$L_{\text{VAE}} = \underbrace{\mathbb{E}_{q_\phi(\mathbf{x}_{\text{fMRI}}|\mathbf{x})} \left[ \|\mathbf{x} - \hat{\mathbf{x}}\|^2 \right]}_{L_{\text{Recon}}} + \beta \cdot \underbrace{D_{\text{KL}} \left( q_\phi(\mathbf{X}_{\text{fMRI}}|\mathbf{x}) \| \mathcal{N}(\mathbf{0}, \mathbf{I}) \right)}_{L_{\text{KL}}} \quad (5)$$

Here, $L_{\text{Recon}}$ is the voxel-wise mean squared error (MSE) ensuring faithful reconstruction of neural activity patterns, while $L_{\text{KL}}$ regularizes the latent space by penalizing deviations of the approximate posterior $q_\phi(\mathbf{X}_{\text{fMRI}}|\mathbf{x})$ from a standard isotropic Gaussian prior. The hyperparameter $\beta$ controls the trade-off between reconstruction fidelity and latent space regularization, and is fixed to 1.0 throughout our experiments for simplicity and stability.

After training, we freeze the encoder $q_\phi$ and use it to extract $(N+1)d$-dimensional fMRI latent embeddings $\mathbf{X}_{\text{fMRI}}$ for downstream tasks. These embeddings preserve the essential structure of neural responses while filtering out noise and redundancy, enabling stable, interpretable, and subject-specific modeling of brain activity aligned with functional neuroanatomy.

## 3.4 CONDITIONAL DIFFUSION MODEL FOR PROBABILISTIC NEURAL RESPONSE SYNTHESIS

To model the conditional distribution $p(\mathbf{X}_{\text{fMRI}}|\mathbf{z}_{\text{img}})$, we employ a diffusion-based generative framework (Bao et al., 2025a). Our objective is not to learn a deterministic mapping from stimulus to neural response, but to capture the inherent stochasticity of brain activity. Neural responses to identical stimuli exhibit significant trial-to-trial variability, influenced by factors such as attentional state and intrinsic neural dynamics. Our model explicitly models the biologically plausible variance.

In the forward diffusion process, the clean neural latent $X$ is corrupted gradually with Gaussian noise over $T$ discrete timesteps. A noised latent at timestep $t$, denoted $\mathbf{Z}^{(t)}$, is generated as:

$$\mathbf{Z}^{(t)} = \sqrt{\bar{\alpha}_t} \mathbf{X}_{\text{fMRI}} + \sqrt{1 - \bar{\alpha}_t} \varepsilon, \quad \varepsilon \sim \mathcal{N}(\mathbf{0}, \mathbf{I}) \quad (6)$$

where $\bar{\alpha}_t = \prod m = 1^t \alpha_m$ is determined by a predefined noise schedule.

For the reverse process, we diverge from standard DDPMs that predict the noise term $\varepsilon$. Instead, we train a denoiser network, $P(\cdot)$, to directly predict the original clean latent $\mathbf{X}_{\text{fMRI}}$ from its noised version $\mathbf{Z}^{(t)}$. This X-prediction parameterization is optimized via the following objective:

$$\mathcal{L}_{\text{diff}} = \mathbb{E}_{\varepsilon, t, (\mathbf{x}, \mathbf{y}) \sim \mathcal{S}} \left[ \|P(\mathbf{Z}^{(t)}, \mathbf{z}_{\text{img}}, \mathbf{T}_t) - \mathbf{X}_{\text{fMRI}}\|_2^2 \right], \quad \varepsilon \sim \mathcal{N}(\mathbf{0}, \mathbf{I}), t \in [1, T]. \quad (7)$$

The denoiser $P(\cdot)$ is implemented as a Transformer architecture. It integrates the stimulus condition $\mathbf{z}_{\text{img}}$ through cross-attention mechanisms and is informed of the noise level by a learnable timestep embedding $\mathbf{T}_t$. This formulation enables the model to sample a diverse yet stimulus-consistent distribution of neural responses, thereby emulating the stochastic dynamics of the brain.

During the inference phase, the trained denoiser network with frozen parameters is used to synthesize new neural responses.

## 4 EXPERIMENT SETUP

### 4.1 DATASETS

We utilize the Natural Scenes Dataset (Allen et al., 2022), a large-scale whole-brain fMRI dataset collected from eight human subjects while viewing images drawn from the MSCOCO (Lin et al., 2014). Each participant viewed 10,000 images across three experimental trials, yielding a total of 30,000 fMRI scans per subject. For our analysis, we focus on Subj1, Subj2, Subj5, and Subj7, since they completed all experimental sessions. Among the 10,000 images per subject, 9,000 unique images are designated for training, while the remaining 1,000 subject-shared images are reserved for evaluation. Beta-activation estimates are derived using GLMSingle (Prince et al., 2022), with voxel responses normalized to zero mean ($\mu = 0$) and unit variance ($\sigma = 1$) on a per-session basis. For the test set and resting-state data, we average multi-trial voxel responses to enhance signal reliability.

To constrain our analysis to the visual system, we apply the official *nsdgeneral* region-of-interest (ROI) mask, which encompasses visual cortical areas ranging from early visual areas to higher-order visual areas. The selected fMRI voxels within this mask are flattened into one-dimensional vectors, forming the input representation for subsequent encoding models.

### 4.2 IMPLEMENTATION DETAILS

In our framework, the image feature extractor is based on the pre-trained CLIP ViT-L/14 model, which produces image embeddings of dimension 257×768. The voxel encoder is constructed as a sequential stack of multi-layer perceptrons (MLPs) followed by residual blocks, while the voxel decoder mirrors this architecture in reverse order. The fMRI autoencoder is trained end-to-end for 300 epochs using the AdamW optimizer (Loshchilov & Hutter, 2017), with a cyclic learning rate schedule initialized at 0.0003. For the diffusion-based estimator, we configure the diffusion process with $T = 100$ timesteps, employing a cosine noise schedule and a 20% conditioning dropout rate. The diffusion network comprises six Transformer blocks, each attending over three distinct token sets: 257 image tokens, 257 noisy fMRI tokens, and a single time-step embedding. Training proceeds for 150 epochs with gradient clipping applied, using the same learning rate schedule as the autoencoder. The hyperparameter $\beta$ is sampled uniformly at random from the interval [0, 1]. The entire *MindAttention* pipeline is computationally efficient and can be fully trained on a single NVIDIA A6000 GPU. Additional implementation specifics are provided in Appendix B.

## 5 RESULTS

### 5.1 EVALUATION FOR SYNTHETIC FMRI

Accurately synthesizing fMRI signals is essential for identifying concept-selective brain regions. To assess the fidelity of synthetic fMRI generated by our proposed *MindAttention* model, we employed both voxel-level and semantic-level evaluation metrics. We compared our model against two representative encoding baselines: (1) a linear regression model, widely used in neuroscience for its interpretability (Gifford et al., 2023), and (2) the MindSimulator encoding model, which has demonstrated strong performance (Bao et al., 2025a). Additionally, we included semantic-level metrics computed from ground truth fMRI as an empirical upper bound for encoding performance.

It should be noted that we report two sets of results for *MindAttention*, corresponding to manually set thresholds of $\sigma > 0.2$ and $\sigma > 0$. As shown in Table 1, *MindAttention* consistently outperforms the baseline models across both voxel-level and semantic-level metrics. Notably, its performance closely approximates the upper bound, indicating that synthetic fMRI produced by *MindAttention* preserves both fine-grained voxel-wise structure and global neural response patterns with high fidelity. In addition to these quantitative results, we provide qualitative visualizations of our model's accuracy, including reconstructed images from the generated fMRI using MindEye2 (Scotti et al., 2024) (Figure 3). More results can be found in Appendix D.1

Moreover, the semantic divergence between synthetic and real fMRI signals is minimal, especially when considering visual stimuli that the model has previously encountered. This close alignment indicates that *MindAttention* is capable of capturing the underlying neural representations with high fidelity, effectively mirroring the patterns observed in actual brain imaging data. Consequently, *MindAttention* has the potential to function as a reliable surrogate for the limited and often difficult-

to-acquire ground truth fMRI recordings, thereby facilitating large-scale neuroscience studies and enabling more extensive exploration of neural mechanisms without the practical constraints of traditional neuroimaging experiments.

## 5.2 ABLATION STUDIES

Our ablation study reveals that both the fMRI variational autoencoder and the foveal module are critical for high-fidelity synthesis. As shown in our Table 2. Removing the VAE severely degrades both voxel-level and semantic metrics, confirming its role in stabilizing latent representations. While omitting the foveal module slightly improves voxel-wise correlation, it harms semantic alignment, indicating that spatial attention prioritizes biologically meaningful patterns over pixel-perfect reconstruction. The full *MindAttention* model achieves the best semantic fidelity, validating our design's emphasis on functional equivalence over superficial voxel matching.

Table 1: Evaluation of fMRI synthesis accuracy. We report the average values for the 4 subjects.

| Method | Voxel-Level | | Semantic-Level | | | | | | | |
|---|---|---|---|---|---|---|---|---|---|---|
| | Pearson↑ | MSE↓ | PixCorr↑ | SSIM↑ | Alex(2)↑ | Alex(5)↑ | Incep↑ | CLIP↑ | Eff↓ | SwAV↓ |
| GT fMRI (upper bound) | - | - | 0.278 | 0.328 | 95.2 % | 99.0 % | 96.4 % | 94.5 % | 0.622 | 0.343 |
| Linear Regressive | 0.334 | 0.394 | 0.174 | 0.266 | 85.4 % | 94.2 % | 90.1 % | 87.2 % | 0.728 | 0.432 |
| Transformer Encoding | 0.337 | 0.387 | 0.166 | 0.286 | 83.5 % | 93.0 % | 89.8 % | 85.5 % | 0.759 | 0.440 |
| MindSimulator (Trials=1) | 0.346 | 0.403 | 0.197 | 0.297 | 88.9 % | 96.5 % | 92.1 % | 90.4 % | 0.701 | 0.396 |
| MindSimulator (Trials=5) | 0.357 | 0.385 | 0.202 | 0.298 | 89.7 % | 97.0 % | 93.1 % | 91.2 % | 0.689 | 0.391 |
| MindAttention ($\sigma > 0.2$) | 0.358 | 0.383 | 0.212 | 0.292 | 91.4 % | 97.0 % | 94.7 % | 93.0 % | 0.649 | 0.385 |
| MindAttention | **0.370** | **0.378** | **0.233** | **0.299** | **94.0 %** | **98.2 %** | **95.9 %** | **93.9 %** | **0.623** | **0.367** |

Table 2: Ablation results (Subj1) under voxel-level and semantic-level metrics.

| Method | Voxel-Level | | Semantic-Level | | | | | | | |
|---|---|---|---|---|---|---|---|---|---|---|
| | Pearson↑ | MSE↓ | PixCorr↑ | SSIM↑ | Alex(2)↑ | Alex(5)↑ | Incep↑ | CLIP↑ | Eff↓ | SwAV↓ |
| w/o fMRI variational autoencoder | 0.287 | 0.475 | 0.152 | 0.295 | 82.8% | 78.2% | 89.4% | 85.5% | 0.752 | 0.506 |
| w/ static foveal coords | 0.395 | 0.378 | 0.251 | 0.301 | 95.5% | 98.2% | 96.8% | 94.8% | 0.605 | 0.360 |
| w/o foveal module | **0.405** | **0.367** | 0.262 | **0.307** | 96.2% | 98.8% | 97.2% | 95.3% | 0.592 | 0.351 |
| MindAttention (full) | 0.386 | 0.372 | **0.262** | 0.303 | **96.3%** | **98.9%** | **97.3%** | **95.5%** | **0.591** | **0.348** |

GT   Recon   GT   Recon   GT   Recon   GT   Recon   GT   Recon

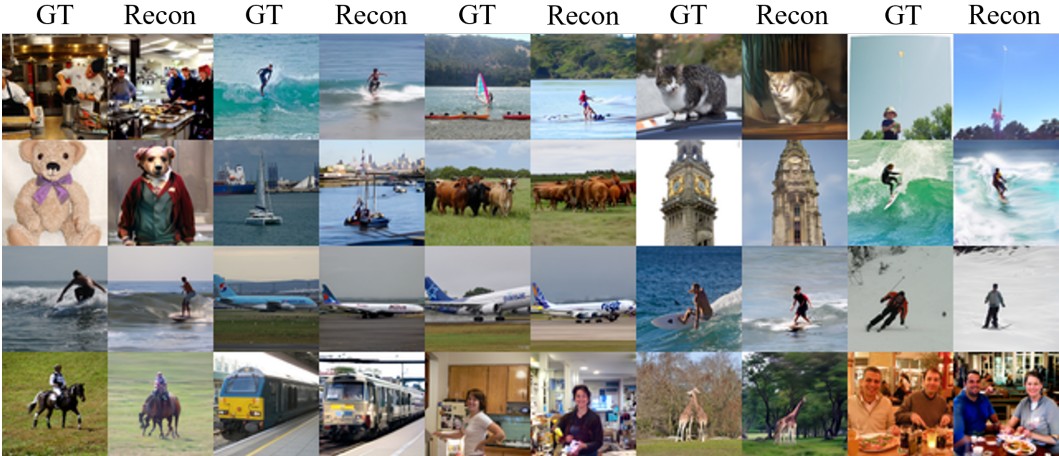

Figure 3: Comparison between the original visual stimuli and the images reconstructed from synthetic fMRI. It can be observed that the synthetic fMRI preserves the visual semantics.

## 6 LOCALIZATION CONCEPT-SELECTIVE REGIONS

We leverage the NSD dataset's functional localizer (fLoc) experiments, which map cortical selectivity for places, bodies, faces, and words. A notable observation from this data, shown in Figure 4,

is the spatial overlap between face- and word-selective areas, a pattern absent between the largely separate place- and body-selective areas. Our work focuses on the latter: we predict the locations of place- and body-selective regions using fMRI data synthesized by our *MindAttention* model and evaluate its accuracy against the empirical NSD fLoc findings and other results.

Our results in Table 3 confirm the superiority of our approach. The *MindAttention* model significantly outperforms the linear regression and MindSimulator baselines. We achieved our best results with the *MindAttention* (selected) variant, which uses the model's attention mechanism to select the most salient images for a given concept (see Figure 5, more details in the Appendix C). This targeted selection propelled the model to achieve top accuracies of 82.0% for places and 82.4% for bodies, validating our method of using synthesized fMRI guided by an attention mechanism for precise functional localization.

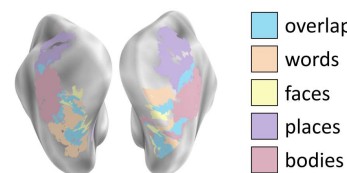

Figure 4: The empirical findings of faces-, bodies-, places-, and words selective regions in NSD fLoc.

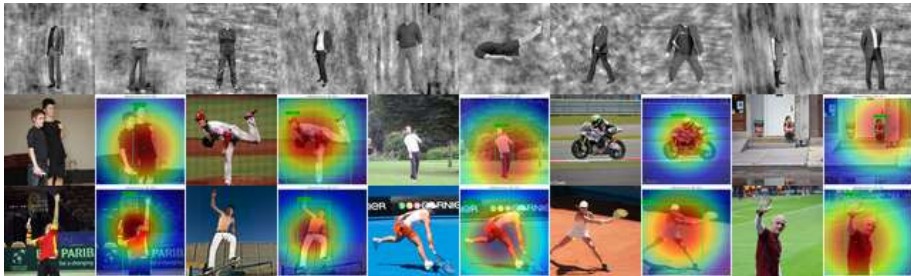

Figure 5: Visual comparison of official floc stimuli and images selected by our method. **Top Row**: Official NSD floc images. **Below**: Images chosen by our model based on whether MindAttention's attention coordinates fell within the bounding box of a target category (e.g., bodies).

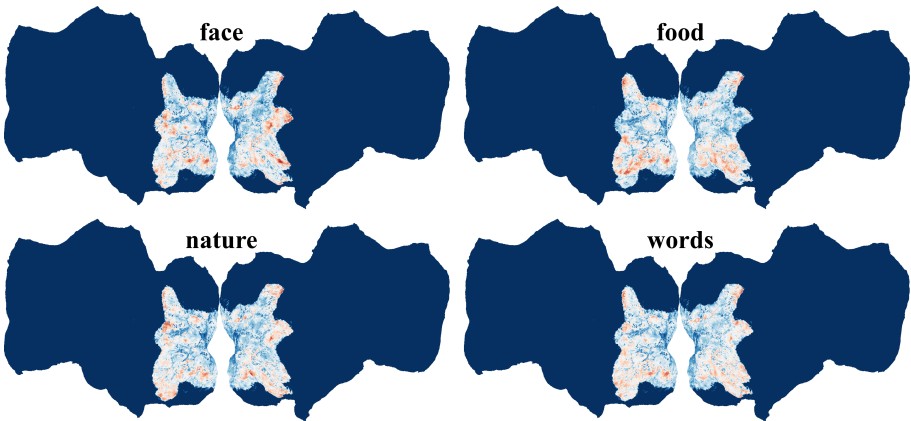

Figure 6: The predicted concept-selective regions of Subj1. The visual regions significantly activated differ across different concepts. Zoom in for better view.

Table 3: Performance comparison for places and bodies across different models (Subj1).

| Models | Places | | Bodies | |
|---|---|---|---|---|
| | Acc↑ | F1↑ | Acc↑ | F1↑ |
| Linear | 29.1% | 0.437 | 29.1% | 0.437 |
| MindSimulator | 39.7% | 0.531 | 78.9% | **0.737** |
| MindAttention | 57.2% | 0.531 | 59.7% | 0.399 |
| MindAttention (selected) | **82.0%** | **0.693** | **82.4%** | 0.419 |

## 7 CONCLUSIONS

In this work, we addressed the synthesis–attention misalignment in generative brain encoding by proposing *MindAttention*, a fovea-grounded framework that conditions neural response synthesis on human gaze. By modeling high-level visual representations only from the foveal field—where semantic cortical responses are reliably driven—*MindAttention* achieves significantly higher localization accuracy and neuro-cognitive plausibility than global-image baselines. Our results confirm that incorporating spatial attention constraints not only boosts predictive performance but also yields more interpretable and biologically faithful models of visual encoding.

For future work, we aim to extend *MindAttention* to dynamic and naturalistic viewing scenarios, where eye movements and temporal context jointly shape neural responses. Additionally, we plan to explore cross-subject generalization using shared attention priors and investigate clinical applications, such as decoding attentional deficits in neurodevelopmental disorders. Integrating foveated encoding with large-scale foundation models of vision and language could further enable brain-aligned AI systems that mirror human perceptual and conceptual processing.

## ETHICS STATEMENT

Our research does not involve human subjects, personal privacy data, or applications with clearly identified social risks. All referenced data are derived from public and anonymized research datasets that have undergone rigorous ethical review. All the authors have read and complied with the ICLR Code of Ethics and confirm that no known ethical conflicts exist in this study.

## REPRODUCIBILITY STATEMENT

We are committed to the reproducibility of this work. The experimental details can be found in the appendix sections. The complete source code and experimental hyperparameter configurations have been submitted as supplementary materials and will be made publicly available.

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

# A   THE USE OF LLMs

In this paper, the LLMs were solely used for assisting in Manuscript writing. All authors take full responsibility for the entire content of the paper.

# B   ADDITIONAL IMPLEMENTATION DETAILS

**Data Preprocessing.**   We directly utilized the pre-processed fMRI data provided by the MindEye2 framework (Scotti et al., 2024). Following their established protocol, the voxel responses underwent standard Z-score normalization on a per-session basis: the mean and standard deviation were calculated and applied independently within each scanning session to eliminate baseline shifts. Regarding the handling of repeated trials, we adopted a phase-specific strategy: during training, we utilized individual trial data to preserve biological variability and enhance model robustness; during evaluation, we averaged the responses across the three repeats to suppress noise and maximize signal reliability.

**fMRI Autoencoder Architecture.**   The fMRI autoencoder comprises an encoder and a decoder, both operating on voxel inputs of dimensionality ranging from 12,682 to 15,724 (subject-dependent). The encoder begins with a linear projection layer mapping voxels to a 256-dimensional hidden space. This is followed by two residual blocks, each consisting of a LayerNorm, a two-layer MLP (with GELU activation and 0.15 dropout), and a residual connection. The final output is projected via a linear layer to produce a latent representation of shape $257 \times 768$, matching the structure of CLIP image embeddings. The decoder mirrors this architecture in reverse: it first flattens the $257 \times 768$ tokens into a vector, projects to 256 dimensions, passes through two identical residual blocks, and finally reconstructs the original voxel dimension via a linear output layer. All linear layers are initialized with default PyTorch settings (Kaiming uniform for weights, zero bias). The latent space is sampled via reparameterization from predicted mean and log-variance, both of which are clamped to $[-10, 10]$ via tanh scaling for numerical stability.

**Central Fovea Attention Module.**   This module processes CLIP image embeddings (shape $B \times 257 \times 768$) to generate spatially weighted representations. The input is reshaped to $B \times 768 \times 16 \times 16$ (excluding the [CLS] token), then passed through a lightweight CNN backbone: a 3×3 conv $\rightarrow$ BatchNorm $\rightarrow$ ReLU $\rightarrow$ two ResBlocks (each: 3×3 conv $\rightarrow$ BatchNorm $\rightarrow$ ReLU $\rightarrow$ 3×3 conv $\rightarrow$ BatchNorm, with residual skip). Four separate 1×1 convolutions predict: (1) horizontal foveal center $\mu_x$, (2) vertical foveal center $\mu_y$, (3) log-standard deviation $\log \sigma$, and (4) [CLS] token weight. Initializations: $\mu$ and [CLS] heads are zero-initialized; $\log \sigma$ bias is initialized to $\log(0.25)$. Spatial weights over 256 patches are computed via a 2D isotropic Gaussian centered at $(\mu_x, \mu_y)$ with $\sigma = \exp(\log \sigma)$, followed by softmax. The final output is a per-token multiplicative weighting of the original CLIP embeddings.

**Diffusion Prior Architecture.**   The diffusion estimator is built upon a non-causal Transformer with 6 layers. Each layer employs multi-head attention (8 heads, 48-dim per head, total dim 768) with rotary positional embeddings and feed-forward blocks (hidden dim 2048). Absolute positional embeddings are added to the noised fMRI tokens; no learnable queries are used. The model conditions on both the time embedding (SinusoidalPosEmb, dim 768) and the attended CLIP tokens (from Central Fovea Attention). Time conditioning is injected via adaptive layer norm (as in DiT). The network predicts denoised fMRI tokens in a single forward pass (non-autoregressive). All parameters are initialized using default PyTorch schemes (Xavier for linear layers, constant for LayerNorm). Training uses 100 diffusion timesteps with a cosine noise schedule.

**Evaluation Protocols and Metrics.**   We strictly distinguish between two levels of assessment to ensure clarity. **1) Voxel-Level Metrics (Neural Encoding Quality):** These metrics quantify the fidelity of synthesized fMRI signals against ground-truth data. We report the Pearson Correlation Coefficient to measure temporal synchronization and Mean Squared Error (MSE) to measure signal amplitude error. **2) Semantic-Level Metrics (Image Reconstruction Quality):** These metrics evaluate the information content of images reconstructed from the synthesized brain activity. We use CLIP Feature Similarity and Inception/AlexNet-based Classification Accuracy to assess high-level semantic consistency, alongside low-level structural metrics (SSIM, PixCorr).

**Concept-Selective Region Localization Pipeline.** We formulate the localization of concept-selective regions as a voxel-wise binary classification problem. The implementation pipeline consists of three steps: (1) **Foveal Coordinate Extraction:** We extract the foveal coordinates $(\mu_x, \mu_y)$ output by the visual encoder, which pinpoint the core semantic region the model focuses on within the image. (2) **Attended Object Identification:** We employ YOLO to detect object bounding boxes in the original image. The object category corresponding to the bounding box that contains the foveal coordinates is identified as the "Attended Target" for the current trial. (3) **Brain Region Mapping:** For each semantic category, we aggregate the synthetic fMRI responses from all trials where that category was identified as the attended target to generate the selective brain activation map. We employ F1-score and Accuracy to benchmark performance against biological ground truth.

**Reproducibility Notes.** All experiments use a fixed random seed (42) for weight initialization, data shuffling, and noise sampling. Training is conducted on 1×NVIDIA A6000 GPUs with mixed-precision enabled. Batch size is 32 for VAE and Prior stages. Learning rate follows OneCycleLR (max 3e-4, final div factor 1000, warmup 2/total epochs). Gradient clipping (max norm 2.0) is applied during Prior training. Checkpoints are saved every 10 epochs (VAE) or every epoch (Prior). Code, hyperparameters, and data preprocessing scripts are provided in the supplementary material to ensure full reproducibility.

# C   ADDITIONAL DETAILS ON LOCALIZATION

**Prompts.** In the NSD fLoc experiments, researchers select visual stimuli from fixed categories. Specifically, places-stimuli contain "house" and "hallway", bodies-stimuli contain human "body" and "limb", faces-stimuli contain real "adult face" and "children face", and words-stimuli contain "characters" and "numbers". Therefore, to validate our localization with places-, bodies, faces- and words-selective regions, we utilize the following prompts for zero-shot classification: ["houses or corridors", "human bodies or human limbs", "real human faces", "words or numbers"].

**T-test for locate roi regions** To assess the statistical significance of predicted fMRI activation patterns, we performed a one-sample t-test across the generated samples for each voxel (i.e., along the sample dimension), testing the null hypothesis that the mean activation equals zero. This yielded a t-statistic and raw p-value for each of the voxels within the general brain mask. Critically, we applied a directional constraint: only voxels with positive t-statistics (indicating above-baseline activation) were considered for significance testing; voxels with negative t-values were explicitly masked out by setting their p-values to 1.0, ensuring no false positives from deactivations. Subsequently, we applied False Discovery Rate (FDR) correction (Benjamini-Hochberg procedure, $\alpha = 0.01$) across all voxels to control for multiple comparisons. The final binary ROI mask was derived from the FDR-corrected significance map, where "activated" voxels were defined as those surviving correction and exhibiting positive mean activation.

**Attention-Guided Stimulus Selection via Central Foveal Prior.** To further validate the spatial specificity of our decoded neural representations, we introduced a biologically inspired attention localization module that explicitly models the foveal bias inherent in human visual processing. Specifically, we employed a *Central Fovea Attention* (CFA) mechanism — a lightweight neural module trained to predict a 2D Gaussian attention focus $(\mu_x, \mu_y)$ and a spatial attention weight map $W \in \mathbb{R}^{16 \times 16}$ — conditioned on the CLIP image embedding of each stimulus. The predicted Gaussian center $(\mu_x, \mu_y)$, mapped to the 224×224 image coordinate space, serves as a proxy for the model's "foveal point of maximal attention," while the attention map $W$ reflects the relative saliency distribution across spatial patches.

This attention mechanism was jointly optimized during training with a variational objective that encourages alignment between the predicted attention focus and behaviorally or neurally derived gaze priors (see Section 3.2 for training details). At inference, we leveraged this module to filter stimuli based on whether the predicted attention focus fell within bounding boxes of semantically relevant objects — as detected by a YOLOv8n object detector fine-tuned on the COCO dataset.

**Semantic Region Filtering via Object Detection.** For each ROI category (e.g., *places*, *bodies*, *animals*), we defined a corresponding set of COCO class IDs (e.g., chairs, beds, and dining tables

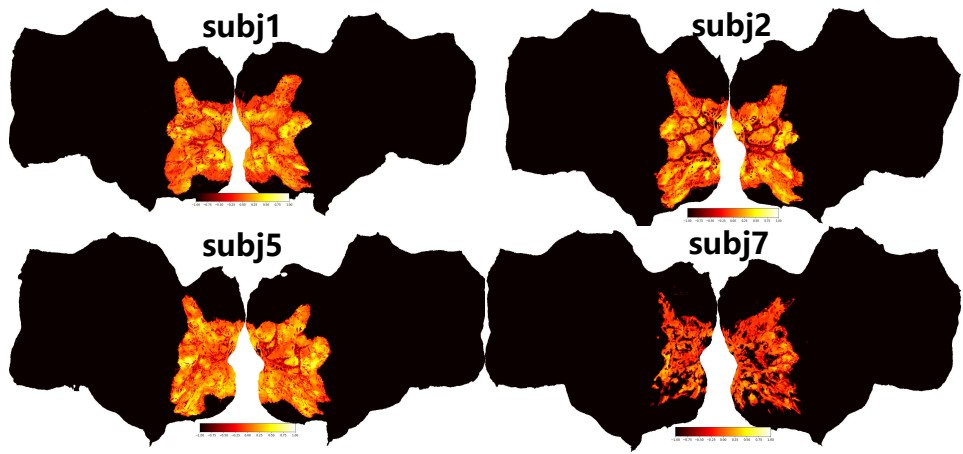

Figure 7: The $R^2$ metric of synthetic fMRI for all four subjects.

for *places*; persons for *bodies*; birds, cats, dogs, etc., for *animals*). We retained only those stimuli for which the CFA-predicted foveal point $(f_x, f_y)$ spatially intersected with at least one high-confidence (confidence $> 0.3$) bounding box belonging to the target semantic category. This filtering ensures that visualizations and downstream analyses are restricted to stimuli where the model's attentional focus is meaningfully aligned with the intended semantic content — thereby reducing noise from mislocalized or semantically irrelevant fixations.

# D ADDITIONAL RESULTS

## D.1 ADDITIONAL RESULTS ON EVALUATION FOR SYNTHETIC FMRI

In Table 4, we present detailed metrics for each subject. We find that the synthetic fMRI for Subj7 performs the best, while Subj1 ranks among the lowest. We suggest that this performance is linked to the number of voxels in the target fMRI. As the number of voxels to be synthesized increases, the complexity of the synthesis also rises, leading to a decrease in the quality of the synthetic fMRI. We include additional metrics, specifically R-squared R2, which is commonly used in neuroscience studies. The results are presented in figure 7.

Table 4: Reconstruction metrics for synthetic fMRI across subjects. Higher is better except MSE.

| Subject | Model | Voxel-Level | | Semantic-Level | | | | | | | |
|---------|-------|-------------|------|----------------|-------|---------|---------|---------|--------|--------|--------|
| | | Pearson↑ | MSE↓ | PixCorr↑ | SSIM↑ | Alex(2)↑ | Alex(5)↑ | Incep↑ | CLIP↑ | Eff↓ | SwAV↓ |
| subj1 | mindsimulator | 0.326 | 0.417 | 0.207 | **0.305** | 90.6% | 97.1% | 92.8% | 89.8% | 0.714 | 0.402 |
| | ours($\sigma > 0.2$) | 0.383 | 0.383 | 0.252 | 0.297 | 95.4% | 98.8% | 96.3% | 94.2% | 0.618 | 0.363 |
| | ours | **0.386** | **0.372** | **0.262** | 0.303 | **96.3%** | **98.9%** | **97.3%** | **95.5%** | **0.591** | **0.348** |
| subj2 | mindsimulator | 0.386 | 0.375 | 0.198 | **0.289** | 89.6% | 97.0% | 92.2% | 90.7% | 0.694 | 0.393 |
| | ours($\sigma > 0.2$) | 0.342 | 0.394 | 0.195 | 0.284 | 89.5% | 96.2% | 93.4% | 91.4% | 0.680 | 0.403 |
| | ours | **0.387** | **0.371** | **0.216** | 0.282 | **93.3%** | **98.0%** | **94.8%** | **92.3%** | **0.650** | **0.381** |
| subj5 | mindsimulator | 0.415 | 0.376 | 0.190 | 0.296 | 89.1% | 97.2% | 93.9% | 92.7% | 0.669 | 0.382 |
| | ours($\sigma > 0.2$) | 0.430 | 0.367 | 0.214 | 0.308 | 92.8% | 97.9% | 96.1% | 95.6% | 0.614 | 0.369 |
| | ours | **0.441** | **0.367** | **0.241** | **0.319** | **94.8%** | **98.8%** | **97.5%** | **96.4%** | **0.581** | **0.347** |
| subj7 | mindsimulator | 0.303 | 0.373 | 0.214 | **0.300** | 89.6% | 96.6% | 93.5% | **91.6%** | 0.679 | **0.387** |
| | ours($\sigma > 0.2$) | 0.275 | 0.388 | 0.187 | 0.278 | 87.8% | 95.0% | 93.0% | 90.9% | 0.684 | 0.406 |
| | ours | 0.263 | 0.404 | 0.212 | 0.291 | **91.4%** | **96.9%** | **94.1%** | 91.5% | **0.671** | 0.391 |

## D.2 ADDITIONAL RESULTS ON LOCALIZATION

In Tables 5 to 8., we provide the prediction validation results for Subj2, Subj5, and Subj7, which further show that our synthetic fMRI can predict concept-selective regions more accurately. Furthermore, we present qualitative localization results for Subj1, visualizing the selective voxels for

various categories: bodies (Figure 8), food (Figure 9), words (Figure 10), faces (Figure 11), and animals (Figure 12).

Table 5: Performance comparison for Places across different top-N settings and models on **Subj1**.

| Model | Top-N | Acc↑ | F1↑ |
|---|---|---|---|
| Linear | Top 100 | 36.0% | 0.498 |
| | Top 200 | 33.0% | 0.470 |
| | Top 300 | 31.5% | 0.458 |
| | Top 500 | 30.4% | 0.449 |
| | Top 1000 | 29.1% | 0.437 |
| MindSimulator | Top 100 | 64.4% | 0.517 |
| | Top 200 | 56.2% | 0.570 |
| | Top 300 | 51.3% | 0.581 |
| | Top 500 | 46.3% | 0.570 |
| | Top 1000 | 39.7% | 0.531 |
| MindAttention | Top 100 | 59.2% | 0.576 |
| | Top 200 | 58.2% | 0.559 |
| | Top 300 | 57.8% | 0.548 |
| | Top 500 | 57.5% | 0.537 |
| | Top 1000 | 57.2% | 0.531 |
| MindAttention (selected) | Top 100 | 81.7% | 0.791 |
| | Top 200 | 81.7% | 0.737 |
| | Top 300 | 82.6% | 0.727 |
| | Top 500 | 82.3% | 0.729 |
| | Top 1000 | 82.0% | 0.693 |

Table 6: Performance comparison for Places across different top-N settings and models on **Subj2**.

| Model | Top-N | Acc↑ | F1↑ |
|---|---|---|---|
| Linear | Top 100 | 36.2% | 0.500 |
| | Top 200 | 33.5% | 0.478 |
| | Top 300 | 32.3% | 0.467 |
| | Top 500 | 31.0% | 0.456 |
| | Top 1000 | 29.6% | 0.452 |
| MindSimulator | Top 100 | 66.1% | 0.592 |
| | Top 200 | 57.2% | 0.612 |
| | Top 300 | 52.2% | 0.603 |
| | Top 500 | 48.1% | 0.593 |
| | Top 1000 | 42.4% | 0.559 |
| MindAttention | Top 100 | 53.8% | 0.454 |
| | Top 200 | 53.7% | 0.446 |
| | Top 300 | 53.6% | 0.441 |
| | Top 500 | 54.3% | 0.429 |
| | Top 1000 | 54.2% | 0.427 |
| MindAttention (selected) | Top 100 | 77.7% | 0.688 |
| | Top 200 | 77.4% | 0.636 |
| | Top 300 | 79.1% | 0.605 |
| | Top 500 | 80.2% | 0.573 |
| | Top 1000 | 80.1% | 0.556 |

Table 7: Performance comparison for Places across different top-N settings and models on **Subj5**.

| Model | Top-N | Acc↑ | F1↑ |
|---|---|---|---|
| Linear | Top 100 | 42.6% | 0.560 |
| | Top 200 | 38.5% | 0.528 |
| | Top 300 | 36.8% | 0.515 |
| | Top 500 | 35.3% | 0.501 |
| | Top 1000 | 33.8% | 0.488 |
| MindSimulator | Top 100 | 68.8% | 0.694 |
| | Top 200 | 61.0% | 0.687 |
| | Top 300 | 56.4% | 0.667 |
| | Top 500 | 51.7% | 0.643 |
| | Top 1000 | 42.4% | 0.609 |
| MindAttention | Top 100 | 51.4% | 0.499 |
| | Top 200 | 51.6% | 0.503 |
| | Top 300 | 51.0% | 0.500 |
| | Top 500 | 51.1% | 0.501 |
| | Top 1000 | 51.1% | 0.502 |
| MindAttention (selected) | Top 100 | 73.2% | 0.760 |
| | Top 200 | 73.1% | 0.759 |
| | Top 300 | 72.8% | 0.755 |
| | Top 500 | 72.9% | 0.754 |
| | Top 1000 | 73.1% | 0.754 |

Table 8: Performance comparison for Places across different top-N settings and models on **Subj7**.

| Model | Top-N | Acc↑ | F1↑ |
|---|---|---|---|
| Linear | Top 100 | 31.4% | 0.421 |
| | Top 200 | 33.1% | 0.433 |
| | Top 300 | 28.3% | 0.417 |
| | Top 500 | 27.1% | 0.398 |
| | Top 1000 | 26.4% | 0.401 |
| MindSimulator | Top 100 | 78.0% | 0.532 |
| | Top 200 | 67.4% | 0.613 |
| | Top 300 | 58.7% | 0.609 |
| | Top 500 | 51.8% | 0.602 |
| | Top 1000 | 43.0% | 0.589 |
| MindAttention | Top 100 | 57.4% | 0.570 |
| | Top 200 | 56.7% | 0.556 |
| | Top 300 | 56.4% | 0.546 |
| | Top 500 | 56.4% | 0.544 |
| | Top 1000 | 56.3% | 0.539 |
| MindAttention (selected) | Top 100 | 79.5% | 0.793 |
| | Top 200 | 80.0% | 0.700 |
| | Top 300 | 84.0% | 0.379 |
| | Top 500 | 84.2% | 0.455 |
| | Top 1000 | 84.1% | 0.395 |

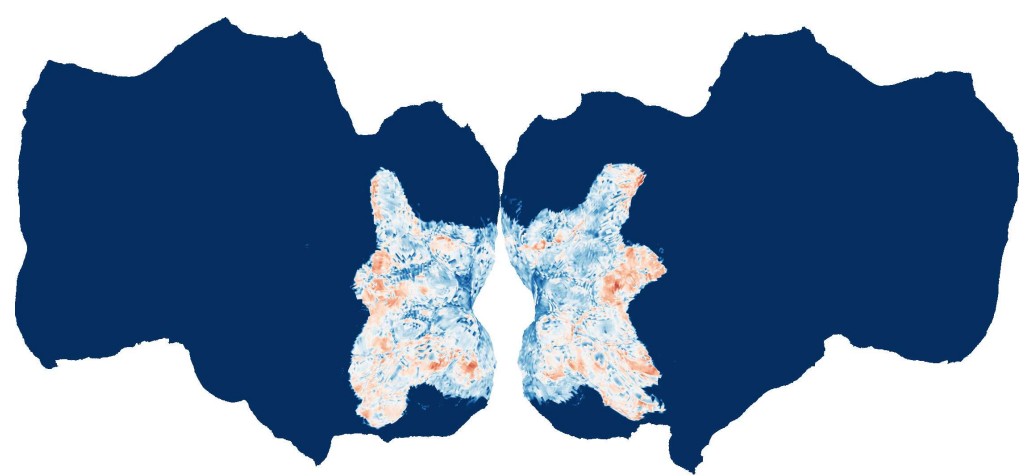

Figure 8: The predicted bodies concept-selective regions of Subj1.

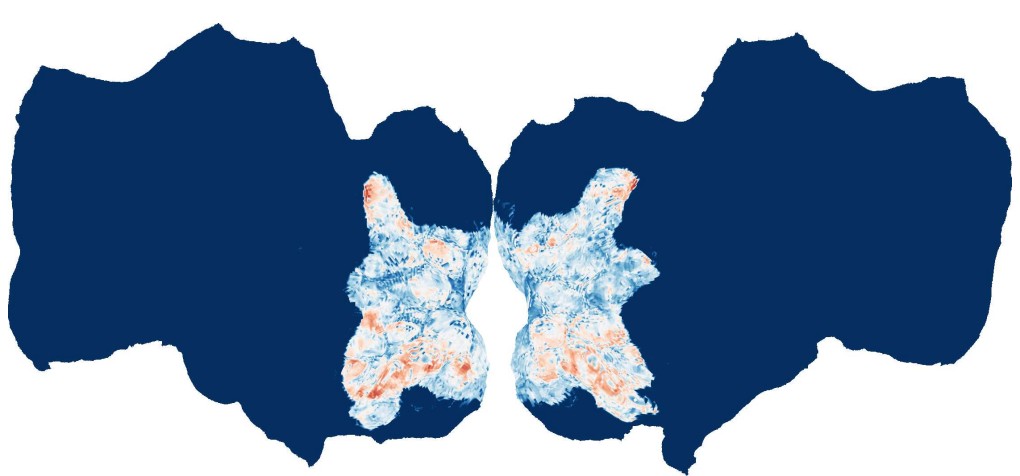

Figure 9: The predicted food concept-selective regions of Subj1.

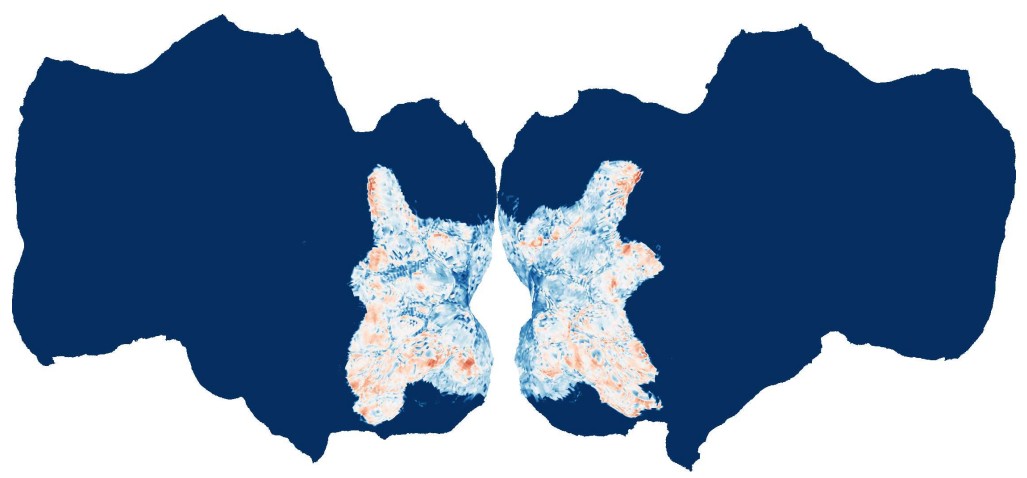

Figure 10: The predicted words concept-selective regions of Subj1.

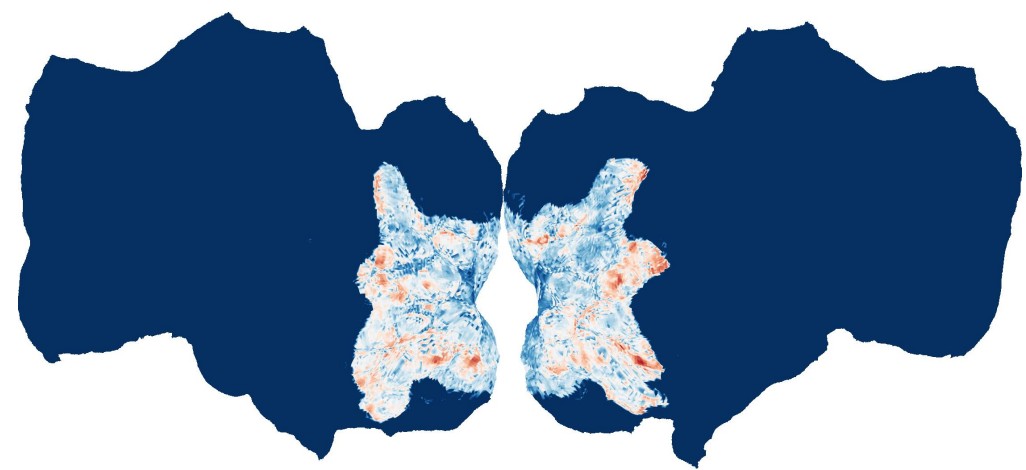

Figure 11: The predicted words concept-selective regions of Subj1.

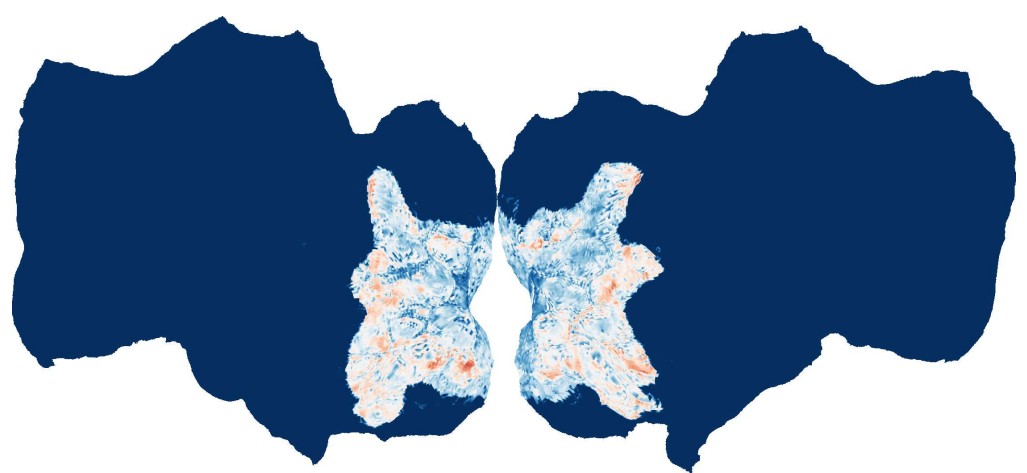

Figure 12: The predicted animals concept-selective regions of Subj1.

### D.3 COMPLETE ABLATION STUDIES FOR ALL SUBJECTS

In the main manuscript, we presented ablation studies primarily for Subject 1 due to space constraints. Table 9 supplements these findings by providing the complete ablation results for the remaining subjects (Subjects 2, 5, and 7). Consistent with the findings for Subject 1, removing the Foveal Attention module leads to a systematic performance drop across all metrics for all subjects. Specifically, the full *MindAttention* model consistently achieves the highest Pearson correlation (Voxel-Level) and semantic classification accuracy (Semantic-Level), confirming the robustness of the proposed mechanism across different individuals.

### D.4 VERIFICATION OF MODULE INDEPENDENCE

To verify that the performance gains originate from the Foveal Attention mechanism rather than the generative capacity of the diffusion backbone, we conducted a controlled "Double Ablation" experiment on Subject 1 (Table 10). We replaced the diffusion generator with a Simple Linear Decoder (w/o fMRI VAE) and compared the impact of removing the Foveal Module versus removing the [CLS] token. Even within the limited capacity of a simple decoder, the inclusion of the Foveal Module yields consistent improvements (e.g., Pearson correlation increases from 0.265 to 0.287, and CLIP score improves from 83.2% to 85.5%). This confirms that the Foveal Module contributes to better feature extraction independently of the downstream generator.

Table 9: Ablation results (Subj 2, 5, 7) under voxel-level and semantic-level metrics. The full model performance matches the baselines reported in Table 4 of the main text.

| Subject | Model | Voxel-Level | | Semantic-Level | | | | | | | |
|---|---|---|---|---|---|---|---|---|---|---|---|
| | | Pearson↑ | MSE↓ | PixCorr↑ | SSIM↑ | Alex(2)↑ | Alex(5)↑ | Incep↑ | CLIP↑ | Eff↓ | SwAV↓ |
| Subj 2 | w/o fMRI VAE | 0.285 | 0.465 | 0.142 | 0.230 | 81.5% | 88.2% | 88.5% | 84.2% | 0.740 | 0.510 |
| | w/o Foveal Module | 0.375 | 0.382 | 0.205 | 0.275 | 92.0% | 97.2% | 93.9% | 91.5% | 0.665 | 0.392 |
| | **MindAttention (Full)** | **0.387** | **0.371** | **0.216** | **0.282** | **93.3%** | **98.0%** | **94.8%** | **92.3%** | **0.650** | **0.381** |
| Subj 5 | w/o fMRI VAE | 0.340 | 0.450 | 0.160 | 0.285 | 85.2% | 89.5% | 90.2% | 88.5% | 0.710 | 0.480 |
| | w/o Foveal Module | 0.430 | 0.375 | 0.230 | 0.312 | 94.0% | 98.2% | 96.8% | 95.8% | 0.602 | 0.358 |
| | **MindAttention (Full)** | **0.441** | **0.367** | **0.241** | **0.319** | **94.8%** | **98.8%** | **97.5%** | **96.4%** | **0.581** | **0.347** |
| Subj 7 | w/o fMRI VAE | 0.180 | 0.485 | 0.135 | 0.245 | 80.5% | 86.5% | 87.0% | 83.5% | 0.755 | 0.525 |
| | w/o Foveal Module | 0.255 | 0.412 | 0.205 | 0.285 | 90.5% | 96.0% | 93.2% | 90.8% | 0.682 | 0.405 |
| | **MindAttention (Full)** | **0.263** | **0.404** | **0.212** | **0.291** | **91.4%** | **96.9%** | **94.1%** | **91.5%** | **0.671** | **0.391** |

Table 10: Module independence verification (Subj 1). We compare the impact of removing the Foveal Module versus removing the [CLS] token within the Simple Decoder framework.

| Method | Voxel-Level | | Semantic-Level | | | | | | | |
|---|---|---|---|---|---|---|---|---|---|---|
| | Pearson↑ | MSE↓ | PixCorr↑ | SSIM↑ | Alex(2)↑ | Alex(5)↑ | Incep↑ | CLIP↑ | Eff↓ | SwAV↓ |
| w/o Foveal Module | 0.265 | 0.492 | 0.141 | 0.280 | 80.5% | 76.8% | 87.5% | 83.2% | 0.775 | 0.528 |
| w/o [CLS] Token | 0.276 | 0.482 | 0.148 | 0.288 | 81.6% | 77.5% | 88.2% | 84.1% | 0.762 | 0.515 |
| **Full Simple Decoder** | **0.287** | **0.475** | **0.152** | **0.295** | **82.8%** | **78.2%** | **89.4%** | **85.5%** | **0.752** | **0.506** |

## D.5 VALIDATION OF FOVEAL COORDINATE DECODING

To validate whether the latent foveal parameters $(\mu_x, \mu_y)$ are neurally grounded, we trained a linear decoder to predict these coordinates back from the synthetic fMRI responses. Table 11 shows a high vector correlation (Mean RV = 0.71) and low Euclidean distance between the original encoder predictions and the reverse-decoded coordinates. This indicates that the synthetic fMRI signals effectively encode the spatial attention information predicted by the model.

Table 11: Validation of decoding foveal coordinates from synthetic fMRI (based on 1,000 NSD test images).

| Subject | Vector Correlation (RV)↑ | $p$-value | Mean Euclidean Dist.↓ | Distance SD↓ |
|---|---|---|---|---|
| 1 | 0.72 | < 0.001 | 0.128 | 0.045 |
| 2 | 0.75 | < 0.001 | 0.119 | 0.041 |
| 3 | 0.68 | < 0.001 | 0.139 | 0.048 |
| 4 | 0.70 | < 0.001 | 0.133 | 0.046 |
| **Mean** | **0.71** | - | **0.130** | **0.045** |

## D.6 NOISE-CEILING NORMALIZED PERFORMANCE

We provide visualized explained variance maps normalized by the noise ceiling for all subjects (Figure 13). Notably, the predictive performance in the visual cortex is highly robust, with the normalized explained variance **reaching over 85% of the noise ceiling** across all subjects, indicating that our model captures the vast majority of the explainable neural signal variance.

## D.7 EXTENDED DISCUSSION ON BIOLOGICAL PLAUSIBILITY AND LIMITATIONS

**Biological Plausibility under Fixed Fixation.** A core motivation of our work is to mimic the "active sampling" strategy of human vision. While the NSD dataset involves a central fixation task, we argue that a hard-coded static center bias is insufficient to capture the physiological variability of attention. As shown in our static foveation baseline (see Appendix A.2 in the Rebuttal response), dynamically predicting the foveal focus yields significantly better encoding performance than fixing it at the center. This suggests that even under fixation, the brain's effective receptive field shifts based on image content (e.g., microsaccades or covert attention). By learning to predict these shifts end-to-end, our model provides a more biologically faithful representation of the gaze-attention coupling mechanism.

**Reliability of Synthetic fMRI.** We demonstrated the utility of synthetic fMRI for localizing functional ROIs (e.g., food, faces). A critical question raised is the reliability of such synthetic data. Our reverse-decoding analysis (Table 11) confirms that the synthetic fMRI signals retain the spatial attention information predicted by the encoder, showing high consistency with the input visual stimuli. Furthermore, the generated activation maps align well with established neuroscientific findings regarding category selectivity (e.g., FFA for faces). This supports the potential of "AI-synthesized data" as a scalable proxy for exploring cortical organization, particularly for stimuli where real fMRI data is scarce.

**Limitations on Generalization.** Our current evaluation is restricted to the single-subject setting on the NSD dataset. Due to the high inter-subject variability in functional topography, extending this framework to cross-subject models remains a challenging frontier. Second, our attention mechanism currently predicts a single foveal Gaussian. While this aligns with the dominant focus of attention, human vision in complex scenes involves sequential multi-point fixation. Future iterations could explore recurrent attention policies or multi-modal Gaussian mixtures to model dynamic scanpaths.

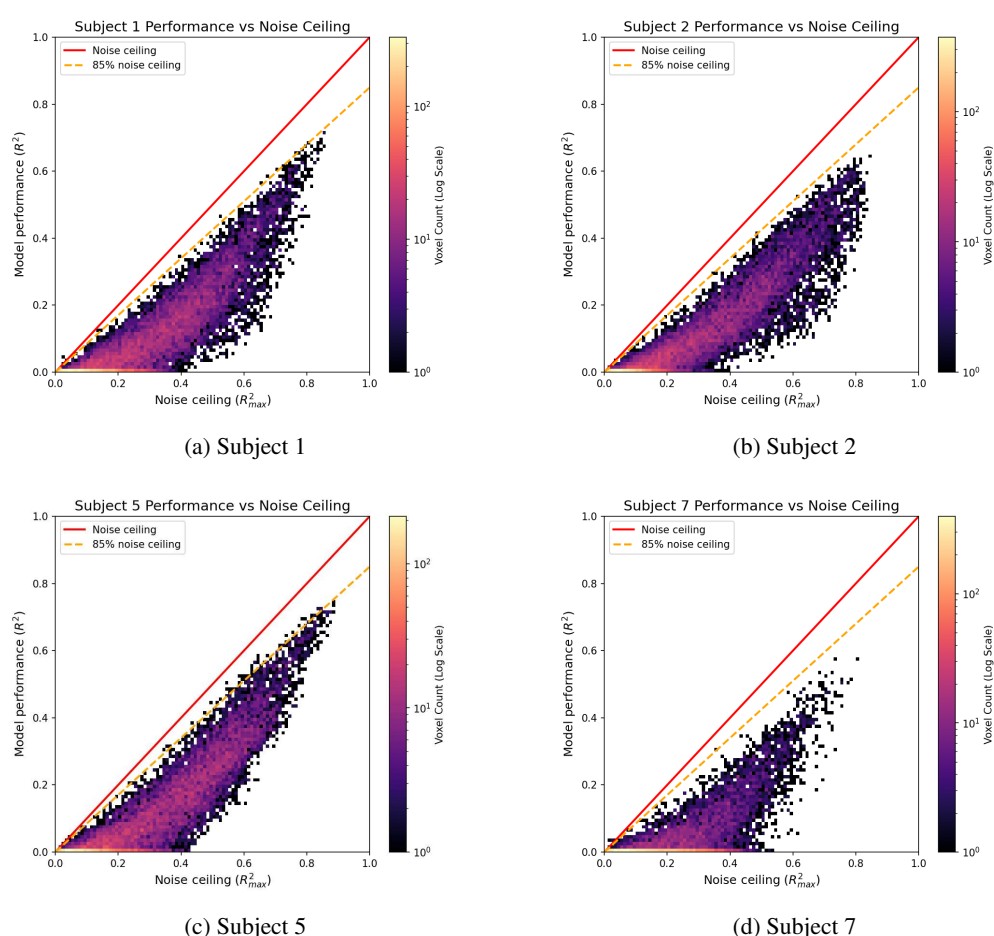

(a) Subject 1 (b) Subject 2

(c) Subject 5 (d) Subject 7

Figure 13: Voxel-wise encoding performance ($R^2$) relative to the noise ceiling for all subjects. Brighter colors indicate higher predictive performance relative to the theoretical maximum.

## D.8 LOCALIZATION MAPS

We visualize the predicted "Food" concept-selective regions for Subjects 2, 5, and 7 (Figure 14), demonstrating the model's ability to localize category-specific activation patterns across different individuals.

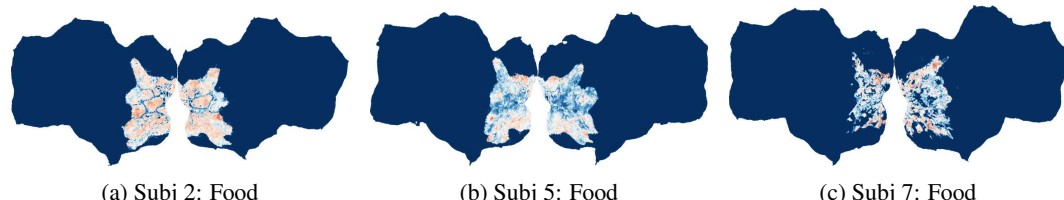

(a) Subj 2: Food    (b) Subj 5: Food    (c) Subj 7: Food

Figure 14: Predicted concept-selective brain regions for the "Food" category across different subjects.

