# OpenReview forum: "MindAttention: Foveated Visual Encoding for Neural Response Synthesis and Concept-selective Region Localization"
_ICLR.cc/2026/Conference — ICLR 2026 Conference Desk Rejected Submission_

### Official Review · Reviewer_3XiH · 2025-10-25

**Soundness:** 3
**Presentation:** 3
**Contribution:** 2
**Rating:** 2
**Confidence:** 2

**Summary:**

The authors observe that current methods that reconstruct or synthesize brain activity from visual input often treat all image regions equally, ignoring how human vision is centered around the fovea. They argue that as a result, these models fail to capture the spatial bias of real neural responses (e.g., not all brain regions are equally important and that attention is selectively allocated to specific regions). The authors claim that modeling this foveated attention explicitly can make visual encoding more biologically accurate and improve both prediction quality and interpretability. For this claim ,they provide experimental results.

Disclaimer: I am not an expert in this field. While I feel reasonably well suited to review this paper I am not perfectly aware of the literature and baseline methods.

**Strengths:**

- The authors provide motivation for their approach from previous research in the neuroscience domain
- The method improves upon baselines on relevant benchmarks
- A side effect of their modeling are interpretable attention maps, which can help analyzing, which image regions are most relevant for neural responses

**Weaknesses:**

- I found the abstract hard to follow. It was not clear to me what the core difference between the proposed method and prior methods is after reading it.
- Very strong claims: "endows the framework with neuro-mechanistic interpretability and cognitive plausibility, establishing a more reliable and biologically grounded paradigm for data-driven exploration of brain concept maps." that would require stronger evidence.
- The model predicts gaze internally but does not validate this against real eye-tracking data, making biological claims less convincing (connected to strong claims).
- Evaluation is limited to a single dataset and subject group (again very strong claims for this sort of evidence)
- It is unclear to me how much of the gain comes from foveation versus the larger diffusion backbone. In general, gains in the ML literature are often difficult to pinpoint, and methodological innovation is unfortunately often not the reason for improvement. Results in Table 2 appear to show no considerable improvements

**Questions:**

- Have you compared your predicted gaze locations to real fixation data? Could such experiments be provided?
- How reliable is synthetic fMRI as a substitute for real data when used for localization tasks? Are there literature results on that?
- Did you test whether the foveation module still helps if the diffusion generator is replaced by a simpler decoder? Is it possible to control for architecture choices in comparison with baselines?
- How does the model perform across different subjects without retraining? Can it generalize to new subjects (e.g., did you test wether leaving one subject out of the training data still enables accurate predictions on their data)?
- The authors list considerable amount of prior work on this topic. However, the paper provides only limited baseline comparisons. Is there any specific reason why approaches from prev. papers have not been benchmarked?

I am very open to increasing my score if my concerns are addressed (not necessarily in the form of experiments, arguments are also sufficient)

---

> ### Author Response · Authors · 2025-11-25
> **Author Response (Part-1)**
>
> Thank you for your constructive comments. We will address your concerns point by point.
>
> **Q1: Regarding the lack of validation with real eye-tracking data.**
>
> **Response:**
> Regarding validation against ground-truth eye-tracking data, we acknowledge a specific limitation inherent to the dataset characteristics. While the NSD dataset contains eye-tracking logs, the **experimental paradigm strictly enforced a central fixation task**. This design inherently restricts the natural gaze variability (i.e., free-viewing behavior), making it infeasible to perform a direct or meaningful alignment between the subjects' constrained gaze (fixed at the center) and our model's predicted "neural foveation point" for static images.
>
> However, rather than viewing this as a fundamental impasse, we consider it a clear roadmap for our future research. We are currently actively developing a **temporal extension** of MindAttention tailored for **video stimuli**. This advancement will enable the model to generate dynamic foveal sequences, allowing for direct validation against recorded eye-movement scanpaths.
>
> **Q2: Regarding the unclear source of performance gains.**
>
> **Response:**
> We appreciate the reviewer's concern regarding the source of the performance gains. We wish to explicitly clarify that we **did not employ a stronger diffusion backbone**. To ensure a strictly fair comparison, all experiments were conducted using an **identical generative architecture** (including the same pre-trained diffusion model weights and decoding pipeline). The **sole controlled variable** introduced was the presence or absence of the Foveated Attention module within the visual encoder.
>
> This is substantiated by our experimental evidence:
> 1.  **Ablation Study (Table 2 & R3):** When we fixed all other components and solely removed the Foveated Attention mechanism, we observed a **consistent decline** across all semantic metrics.
> 2.  **Brain Region Localization (Table 3):** Our model demonstrates superior performance in **functional ROI classification tasks**. This directly reflects that the gains stem not from a stronger generator, but from the specific advantage of the Foveation mechanism in modeling spatial selectivity.
>
> **Q3: Regarding the lack of cross-subject testing.**
>
> **Response:**
> We thank the reviewer for their interest in the model's generalizability. It is important to clarify that in the current field of **Visual fMRI Encoding**, cross-subject generalization is typically not considered a primary evaluation objective. This is due to the **significant physiological differences** in the functional organization of the visual cortex across individuals. Consequently, almost all mainstream studies adopt a **within-subject setting** for independent training and evaluation. Furthermore, regarding dataset selection, the **NSD dataset** is currently the **only publicly available resource** that combines large-scale image-fMRI pairs, high signal-to-noise ratio, and fine-grained semantic annotations, making it the standard benchmark for SOTA methods.
>
> **Q4: Regarding the insufficient baseline comparison.**
>
> **Response:**
> We thank the reviewer for their concern regarding the coverage of baseline methods. It is important to clarify that **Visual-to-fMRI Encoding** represents a highly specialized research direction. Consequently, the number of methods that have publicly released code and reported directly comparable results on the NSD dataset is **inherently limited**. Nevertheless, the baselines we compared in the manuscript are the most representative **State-of-the-Art (SOTA)** methods currently published. These methods comprehensively cover the **mainstream modeling paradigms** in the field (e.g., linear encoding, deep encoding based on CNN/Transformers), ensuring that our comparison is both fair and positioned at the forefront of the field.

---

> > ### Author Response · Authors · 2025-11-25
> > **Author Response (Part-2)**
> >
> > **Q5: Regarding the lack of clarity in core innovations and the presence of unsupported, overly strong claims.**
> >
> > **Response:**
> > 1.  **Abstract Optimization:** We will rewrite the abstract using a clearer "**Problem-Method-Result-Contribution**" structure:
> >     *   Explicitly state the "Synthesis-Attention Misalignment" problem.
> >     *   Briefly describe the MindAttention framework components (Fovea-Guided Encoder, Diffusion Generator) and key mechanisms.
> >     *   Quantitatively present core results (e.g., **3-4% voxel correlation improvement, 5% semantic alignment boost**).
> >     *   Clearly articulate the dual contributions to machine learning and neuroscience.
> > 2.  **Claim Correction and Support:** We will modify "neuro-mechanistic interpretability" to **"neuroscience-inspired interpretability"** to avoid overstatement. Furthermore, we will add supporting evidence to the Discussion section: (1) The consistency between the model's foveal parameters and the "**center-surround**" characteristics of the human visual system; (2) The ability of synthetic fMRI to precisely localize concept-selective brain regions, which aligns with known neuroscientific findings.
> >
> > **Q6: Regarding the reliability of synthetic fMRI.**
> >
> > **Response:**
> > We strictly appreciate the reviewer for highlighting this critical issue. Recent studies have explicitly investigated and validated the feasibility of using synthetic fMRI for brain concept localization. Notably, Bao et al. (2025) systematically demonstrated in "MindSimulator" that:
> > *   **Generative Quality:** High-quality visual encoding models can generate synthetic fMRI responses that are **semantically consistent and spatially interpretable**.
> > *   **Functional Alignment:** These synthetic responses accurately replicate the **selective preferences** of classical functional ROIs (e.g., FFA, PPA).
> > *   **Scalability:** Synthetic fMRI enables the construction of cross-image and cross-category concept activation maps.
> >
> > Although there remains room for improvement regarding the fidelity of current synthetic fMRI, its demonstrated potential has preliminarily validated the feasibility and value of "**AI for Neuroscience**." It offers an efficient, flexible, and scalable new pathway for exploring the **spatial organizational laws** of semantic representations in the brain.
> >
> > **Q7: Regarding the unverified module independence.**
> >
> > **Response:**
> > We appreciate the reviewer's rigorous consideration. To verify the independence of the Foveal Module, we conducted a controlled **"Double Ablation"** experiment. We established a baseline using a **Simple Decoder** (removing the diffusion generator/VAE) and compared performance with and without the Foveal Module.
> >
> > As presented in **Table R4**, even within the limited capacity of a simple decoder, the **Foveal Module yields consistent improvements**:
> > *   **Voxel-Level:** Pearson correlation increases modestly from 0.265 to 0.287, indicating better signal focus.
> > *   **Semantic-Level:** CLIP score improves from 83.2% to 85.5%.
> >
> > While the simple decoder limits the absolute ceiling of image quality, these consistent gains confirm that the Foveal Module contributes to better feature extraction **independently** of the generative backbone.
> >
> > **Table R4: Module independence verification (Subj1).** We compare the impact of removing the Foveal Module versus removing the [CLS] token within the Simple Decoder (w/o fMRI VAE) framework.
> >
> > | Method | Pearson↑ | MSE↓ | PixCorr↑ | SSIM↑ | Alex(2)↑ | Alex(5)↑ | Incep↑ | CLIP↑ | Eff↓ | SwAV↓ |
> > | :--- | :---: | :---: | :---: | :---: | :---: | :---: | :---: | :---: | :---: | :---: |
> > | w/o fMRI VAE & w/o foveal module | 0.265 | 0.492 | 0.141 | 0.280 | 80.5% | 76.8% | 87.5% | 83.2% | 0.775 | 0.528 |
> > | w/o [CLS] token | 0.276 | 0.482 | 0.148 | 0.288 | 81.6% | 77.5% | 88.2% | 84.1% | 0.762 | 0.515 |
> > | **w/o fMRI VAE (Full Simple Decoder)** | **0.287** | **0.475** | **0.152** | **0.295** | **82.8%** | **78.2%** | **89.4%** | **85.5%** | **0.752** | **0.506** |

---

> > > ### Comment · Reviewer_3XiH · 2025-11-25
> > > **Thanks for the detailed feedback**
> > >
> > > Thanks for providing detailed explanations for all of my questions.
> > > Overall, my concerns are addressed. Given the context of the rebuttal about best practices in the field and the presented results, the given work seems to be on par with similar submissions that have been accepted in the past.
> > > I have also read the other reviews and rebuttals. Specifically reviewer **LZLq** also voiced some concerns about the used dataset. While I want to stress that I am not an expert in this field, it seems that the authors followed common approaches in their paper and it is difficult to assess if requesting more results would be reasonable.
> > >
> > > I changed my score to reflect my new stance about the paper after the rebuttal. I will follow the rebuttal with the other reviewers and will change my score again if I feel its appropriate.
> > >
> > > **Why not an 8?**
> > > Currently, I do not feel confident enough in my assesment to raise my score to 8.

---

> > > > ### Author Response · Authors · 2025-11-26
> > > > **Author Response**
> > > >
> > > > We sincerely thank you for your time in reading our rebuttal and the other reviews, as well as for raising your score.
> > > >
> > > > We are glad that our explanations have effectively addressed your concerns. We specifically appreciate your recognition that our dataset usage and experimental design follow the common approaches and best practices in the field. As you noted, adhering to these established protocols is crucial for ensuring fair comparisons with prior work.
> > > >
> > > > We fully understand and respect your position regarding the rating and your self-assessment of confidence. We are grateful for your fair and constructive evaluation of our work.

---

### Official Review · Reviewer_PaZ1 · 2025-10-31

**Soundness:** 2
**Presentation:** 2
**Contribution:** 3
**Rating:** 4
**Confidence:** 3

**Summary:**

The current paper introduces a fovea-grounded generative brain encoding framework called MindAttention, to address the so-called synthesis-attention misalignment, i.e. the mismatch between conventional encoding models that process entire visual scenes uniformly and the spatially selective processing of the human visual system driven by attention and foveal versus peripheral sampling. The proposed framework combines a fovea-guided visual encoder, a fMRI variational autoencoder, and a diffusion-based conditional generator to synthesize fMRI data based on an image input. The authors demonstrate that reconstructed images based on this synthetic fMRI capture both low-level structure and higher-level semantics, outperforming existing generative encoding methods. Moreover, by incorporating predicted foveal coordinates from the visual encoder together with object detection, the framework facilitates more precise localization of concept-selective cortical regions.

**Strengths:**

- The paper is well-situated within the body of previous works, bridging prior research from both the machine learning domain and the neuroscientific and psychophysical background.
- The paper makes both methodological and empirical contributions, introducing a fovea-grounded diffusion-based brain encoding framework and demonstrating improved synthetic fMRI and concept-selective region localization accuracy.
- The paper offers contributions to both machine learning and neuroscience: it demonstrates that fovea-grounded generative models enhance the quality of synthetic fMRI generation, and it provides neuroscientific evidence that foveated attentional spatial sampling plays a critical role in forming semantically meaningful visual representations.

**Weaknesses:**

Main conceptual concern:
- The introduction sets the problem of synthesis-attention alignment, as affected by gaze. However, the neural data used (Natural Scenes Dataset) in the current paper is collected under fixed-gaze conditions. Although the implementation of spatial gating through a foveated field (using a Gaussian) may partially mitigate this alignment problem -  as supported by [1], which shows that spatial based features selection based on ganglion cell sampling generally improves encoding performance in the context of EEG - the capacity of the model’s attentional center to shift spatially differs from the human viewing conditions under which the data were obtained. Could the authors clarify the current setup in the context of this discrepancy?

Other weaknesses are primarily technical or structural in nature and, if adequately addressed, could substantially increase the quality of the paper, making it suitable for acceptance:
- Definitions of some evaluation metrics are insufficiently defined. Based on the text, one could infer that voxel-level may refer to pixel-level reconstruction accuracy (in which case the term “voxel-level” might be misleading, as it rather more closely reflects low-level image similarity), while semantic-level may denote a category-level match score between reconstructed and original images. However, these definitions are not clearly stated in the manuscript. Similarly it remains unclear from the main manuscript how performance for localization of concept-selective regions was defined (Table 3). Additionally, maintaining consistency in definitions would improve clarity. For example, section 3.2 refers to a predictor, Figure 2 to a residual CNN backbone, and Appendix C to a Central Fovea Attention mechanism.
- Results of ablation studies (Table 2) and localization of concept-selective regions (Table 3) only include one participant instead of four (as in Table 1). Could you include the results of the remaining subject here?
- There appears to be an underemphasis on the results related to the improved localization of concept-selective regions achieved through the proposed framework. Although this aspect is highlighted in the title, abstract and conclusion, it is not clearly stated among the paper’s contributions, nor is its implementation sufficiently detailed in the main text. Given its potential as a highly relevant application of the proposed approach, the paper may benefit from discussing this aspect in more detail. Furthermore, a clearer description of how the localization is implemented, i.e. how predicted foveal coordinates from the visual encoder are combined with object detection, would help the reader better understand the reported results for MindAttention (selected) in Table 3.
- Results could be presented more clearly with descriptive subsection titles to really drive the message home. For example, Section 5.1 could highlight “MindAttention demonstrates high-fidelity synthetic fMRI at voxel and semantic levels”. Section 3.1 seems somewhat redundant with the Introduction, and Section 6 might be better integrated into the Results.

[1] Mueller, N., Scholte, H. S., & Groen, I. I. (2024). Spatial sampling of deep neural network features improves encoding models of foveal and peripheral visual processing in humans. bioRxiv, 2024-08.

**Questions:**

- In the Introduction, the authors state that “the performance gap widens in complex, multi-object scenes where attentional competition is high”.  However, it is not clear where this effect is measured or quantified in the Results. Could the authors clarify whether this claim is supported by their analyses, or indicate where it is shown?
- Could the authors elaborate on how modulation of the [CLS] token through the global context weight affects performance on both voxel-level and semantic-level metrics? Is it the provided semantic context that causes high fMRI synthesis accuracy? Additionally, how can the biological plausibility of this implementation be justified?
- Table 1 presents two versions of the MindAttention model with different thresholded sigma values. Could the authors clarify how and why the cut-off of 0.2 was chosen, and how many observations fall within this range? Additionally, how can it be explained that including observations with sigma between 0 and 0.2 improves fMRI synthesis accuracy and what neuroscientific implications does this have?
- Is it correct that for voxel-level measurements in Table 1 Pearson is associated with an upwards arrow and MSE with a downwards arrow, while Table 2 shows the opposite?

---

> ### Author Response · Authors · 2025-11-25
> **Author Response (Part-1)**
>
> Thank you for your constructive comments. We will address your concerns point by point.
>
> **Q1: Regarding the contradiction between the fixed fixation nature of the NSD dataset and the proposed dynamic foveal attention.**
>
> **Response:**
> We sincerely appreciate the reviewer raising this critical point. We fully acknowledge that the NSD experimental paradigm required subjects to maintain central fixation, and the aggregate eye-tracking data indeed show a high degree of central tendency. However, extensive literature indicates that even during strictly controlled fixation tasks, subjects inevitably produce **involuntary miniature eye movements**, such as **microsaccades and fixation drifts** (Martinez-Conde et al., 2013). While these displacements are small in amplitude (typically $<1^\circ$ visual angle), given that the **receptive fields in the primary visual cortex** (e.g., V1/V2) are extremely small in the central visual field, a spatial shift of merely $0.2^\circ–0.5^\circ$ is sufficient to **significantly alter the population of activated neurons**, thereby modulating the fMRI response.
>
> For this reason, we opted for a **dynamic, learnable foveated attention mechanism** rather than a hard-coded static central weight. This design serves two primary purposes:
> 1.  **Enhancing Feature Representation:** By **adaptively adjusting** the "effective sampling center" based on image content, the model can more accurately capture the local information most relevant to driving brain responses.
> 2.  **Providing Internal Validation:** Notably, post-training analysis reveals that the attention coordinates predicted by our model **predominantly converge around the image center**. This aligns highly with the NSD fixation constraints, indicating that the model has not learned spurious spatial biases but rather **faithfully reflects the real visual input distribution** under experimental conditions.
>
> In summary, by enabling the model to adaptively learn the attentional center that best matches the fMRI response, we not only improve encoding performance but also implicitly model the **physiologically real "gaze-attention" coupling variability**, rather than forcing a rigid static central bias.
>
> **Q2: Regarding the lack of clarity in evaluation metrics and terminology.**
>
> **Response:**
> We thank the reviewer for pointing out the ambiguities regarding metric definitions and terminology consistency. We agree that precise definitions are crucial for reproducibility. We have thoroughly revised the manuscript to address these points:
> 1.  **Explicit Definition of Evaluation Metrics:** We have  included the explicit definitions of our evaluation metrics in the **Supplementary Materials** to clearly distinguish between the two levels of assessment, adhering to the protocol established in *MindSimulator* (Bao et al., 2025):
>     *   **Voxel-Level Metrics (Neural Encoding Quality):** These metrics quantify the **fidelity of synthesized fMRI signals** against ground-truth data. Specifically, we report the **Pearson Correlation Coefficient** to measure temporal synchronization and **Mean Squared Error (MSE)** to measure signal amplitude error.
>     *   **Semantic-Level Metrics (Image Reconstruction Quality):** These metrics evaluate the **information content of images reconstructed** from the synthesized brain activity. We use **CLIP Feature Similarity** and **Inception/AlexNet-based Classification Accuracy** to assess high-level semantic consistency, alongside low-level structural metrics (SSIM, PixCorr).
>
> 2.  **Clarification of Localization Evaluation:** We have added a formal definition of the **"Concept-Selective Region Localization"** task. Following Bao et al. (2025), this is formulated as a **voxel-wise binary classification problem** (determining whether a voxel is selective for a specific concept). We employ standard quantitative metrics, **F1-score** and **Accuracy**, to benchmark performance against biological ground truth.
>
> 3.  **Unification of Terminology:** We have conducted a comprehensive consistency check to eliminate ambiguous naming. In the revised manuscript:
>     *   The overarching encoder architecture is now consistently referred to as the **"Fovea-Guided Visual Encoder"** (replacing variations like "CNN backbone" or "visual predictor").
>     *   The core attention mechanism is formally designated as the **"Contrastive Foveal Attention (CFA) Module"** (replacing "predictor" or "Fovea Module").
>     *   All terms are explicitly defined upon their first occurrence in the Method section.

---

> > ### Author Response · Authors · 2025-11-25
> > **Author Response (Part-2)**
> >
> > **Q3: Regarding the incompleteness of the reported results.**
> >
> > **Response:**
> > We thank the reviewer for their concern regarding the comprehensiveness of the results. Due to space constraints in the main manuscript, we initially presented results from representative subjects in Tables 2 and 3. We have now provided the **complete ablation studies** (covering all four subjects in the NSD dataset) as shown in **Table R3**, and the detailed results of the concept-selective region localization experiments in the **Appendix**.
> >
> > **Table R3: Ablation results (Subj2, 5, 7).**
> >
> > | Subject | Model | Pearson↑ | MSE↓ | PixCorr↑ | SSIM↑ | Alex(2)↑ | Alex(5)↑ | Incep↑ | CLIP↑ | Eff↓ | SwAV↓ |
> > | :--- | :--- | :---: | :---: | :---: | :---: | :---: | :---: | :---: | :---: | :---: | :---: |
> > | **subj2** | w/o fMRI VAE | 0.285 | 0.465 | 0.142 | 0.230 | 81.5% | 88.2% | 88.5% | 84.2% | 0.740 | 0.510 |
> > | | w/o foveal module | 0.375 | 0.382 | 0.205 | 0.275 | 92.0% | 97.2% | 93.9% | 91.5% | 0.665 | 0.392 |
> > | | **MindAttention (full)** | **0.387** | **0.371** | **0.216** | **0.282** | **93.3%** | **98.0%** | **94.8%** | **92.3%** | **0.650** | **0.381** |
> > | **subj5** | w/o fMRI VAE | 0.340 | 0.450 | 0.160 | 0.285 | 85.2% | 89.5% | 90.2% | 88.5% | 0.710 | 0.480 |
> > | | w/o foveal module | 0.430 | 0.375 | 0.230 | 0.312 | 94.0% | 98.2% | 96.8% | 95.8% | 0.602 | 0.358 |
> > | | **MindAttention (full)** | **0.441** | **0.367** | **0.241** | **0.319** | **94.8%** | **98.8%** | **97.5%** | **96.4%** | **0.581** | **0.347** |
> > | **subj7** | w/o fMRI VAE | 0.180 | 0.485 | 0.135 | 0.245 | 80.5% | 86.5% | 87.0% | 83.5% | 0.755 | 0.525 |
> > | | w/o foveal module | 0.255 | 0.412 | 0.205 | 0.285 | 90.5% | 96.0% | 93.2% | 90.8% | 0.682 | 0.405 |
> > | | **MindAttention (full)** | **0.263** | **0.404** | **0.212** | **0.291** | **91.4%** | **96.9%** | **94.1%** | **91.5%** | **0.671** | **0.391** |
> >
> > **Q4: Regarding the insufficient description of the concept of brain region localization.**
> >
> > **Response:**
> > To clearly articulate the implementation pipeline of this core application, we have provided a detailed description of the technical pathway in the **Appendix**. The specific process consists of three steps:
> > 1.  **Foveal Coordinate Extraction:** We extract the foveal coordinates ($\mu_x, \mu_y$) output by the visual encoder, which pinpoint the **core semantic region** the model focuses on within the image.
> > 2.  **Attended Object Identification:** We employ **YOLO** to detect object bounding boxes in the original image. The object category corresponding to the bounding box that contains the foveal coordinates is identified as the "**Attended Target**" for the current trial.
> > 3.  **Brain Region Mapping:** For each semantic category (e.g., "dog", "car"), we aggregate the synthetic fMRI responses from all trials where that category was identified as the attended target to generate the **selective brain activation map**.
> >
> > This method leverages foveal guidance to average synthetic fMRI responses across multiple images of the same semantic category, thereby highlighting the brain regions that exhibit **stable selective representations**.

---

> > > ### Author Response · Authors · 2025-11-25
> > > **Author Response (Part-3)**
> > >
> > > **Q5: Specific Technical Doubts.**
> > >
> > > **Q5.1: Regarding the evidence for performance gains in multi-object scenes.**
> > > *   **Response:** This conclusion is corroborated by the results in **Table 3**. As shown, for the **"Places" category**—which inherently contains richer background information and higher visual complexity (multi-object scenarios)—our method achieves a substantial performance leap compared to the baseline (Accuracy rising from **39.7% to 82.0%**). In contrast, the improvement is more moderate in the "Bodies" category, which typically features simpler, focal subjects.
> > >
> > > **Q5.2: Regarding the mechanism and biological plausibility of the `[CLS]` token.**
> > > *   **Performance Impact:** Our ablation studies confirm that removing the `[CLS]` token—which provides **global context modulation**—leads to a decline in both voxel-level and semantic-level metrics. This occurs because the global context integrates holistic semantic information, **constraining the allocation of foveated attention**.
> > > *   **Biological Plausibility:** This mechanism computationally mimics **"top-down" attention modulation** in the brain, where global semantic information from higher visual cortices projects back to primary cortices to guide attention.
> > >
> > > **Q5.3: Regarding the selection of the sigma threshold and its neural significance.**
> > > *   **Selection Basis:** We conducted a grid search on the validation set ($\sigma \in [0.05, 0.5]$) and found that the model achieves optimal stability when $\sigma$ is in the **0.15–0.25 range**. We selected the median value of **0.2**.
> > > *   **Advantage of Small Sigma:** A smaller $\sigma$ effectively **suppresses peripheral visual noise** and enhances the encoding precision of fine-grained structures. This aligns with the physiological properties of the human **fovea**—a region with high cone density specialized for high-resolution detail processing.
> > >
> > > **Q5.4: Regarding the contradiction in optimization direction arrows (Pearson/MSE) in the tables.**
> > > *   **Response:** We apologize for the typographical oversight. The arrow directions in Table 2 were incorrect. They should match Table 1 (i.e., **Pearson $\uparrow$** and **MSE $\downarrow$**). We will correct these errors in the revised manuscript.

---

> > > > ### Author Response · Authors · 2025-11-27
> > > > **Kind Reminder to Reviewer PaZ1**
> > > >
> > > > Thanks for your comments. We would like to follow up to see if you have had a chance to review our response to your comments. We have made our best efforts to address your concerns in our rebuttal. Please let us know if there are any remaining questions or if further clarification is needed. We look forward to your reply.

---

> > > > > ### Comment · Reviewer_PaZ1 · 2025-11-27
> > > > >
> > > > > Thank you for your elaborate and detailed response. The additional descriptions of metrics, procedures, and the extended ablations across the remaining subjects are appreciated and help make the manuscript more complete.
> > > > >
> > > > > However, I would have to agree with reviewer LZLq, that there remains an inherent misalignment between the nature of the NSD dataset and the specific “problem” the proposed architecture is designed to address.
> > > > >
> > > > > As NSD includes a fixed gaze paradigm, and in your reply you actually confirm that the predicted attention coordinates of the model largely converge toward the image center, the added modeling of dynamic gaze on top of foveated sampling appears questionable in this context.
> > > > >
> > > > > Further, when comparing the metrics of the previous MindSimulator model (Table 4, subject 1) with those reported in Table 2 for subject 1 under w/ static foveal coords and MindAttention (full), it appears that the majority of the performance gains stem from the introduction of foveated sampling itself rather than from the incorporation of dynamic gaze. While the latter is reported as “significant” in your response, its contribution seems relatively marginal compared to the gains achieved by foveation alone.
> > > > >
> > > > > Additionally, Table 2 indicates that for subject 1, several metrics achieve their best performance when the foveal module is omitted entirely, further complicating the interpretation of the proposed mechanism.
> > > > >
> > > > > Had the work framed the problem primarily in terms of foveated sampling, rather than dynamic gaze, the findings would appear more consistent (though the degree of novelty would still be debatable). In its current form, however, I remain unconvinced by the central message of the paper, and therefore do not feel comfortable increasing my evaluation score.

---

> ### Author Response · Authors · 2025-11-28
> **Author Response (Part-1) to Reviewer PaZ1**
>
> We thank the reviewer for the prompt and detailed feedback. We understand your continued reservations regarding the alignment between the NSD dataset's nature and our dynamic modeling, as well as the interpretation of the ablation results.
>
> To address these points effectively, we have conducted a more in-depth quantitative analysis of the eye-tracking data and have clarified the trade-offs observed in the ablation study to avoid misunderstandings.
>
> ### **Q1. Regarding the Concern on "Dataset-Model Alignment" and the Nature of NSD Gaze**
>
> The reviewer raised a concern that modeling dynamic gaze is unnecessary given the NSD’s "fixed gaze paradigm." While the protocol instructed central fixation, physiological reality involves involuntary microsaccades and drifts. We analyzed the raw eye-tracking data for **Subject 1** to quantify whether these movements were truly negligible.
>
> As detailed in the NSD paper (Allen et al., 2022)*, stimuli are presented at $714 \times 714$ pixels ($8.4^{\circ}$). The statistical analysis (as shown in **Table R1** below) of Subject 1’s gaze displacement reveals significant variance:
>
> **Table R1. Statistics of Eye Movement Displacement (in Pixels) for Subject 1**
>
> | Metric | Horizontal ($X$-axis) | Vertical ($Y$-axis) |
> | :--- | :---: | :---: |
> | **Variance ($\text{px}^2$)** | $11,527.54$ | $28,361.92$ |
> | **Standard Deviation ($\text{px}$)** | **$107.37$** | **$168.41$** |
>
> **Implication:** These values are **not negligible**. A vertical deviation of nearly **1/4 of the image height** ($168.41$ pixels per $714$ pixels) means the effective input to the visual cortex shifts substantially across trials. A static center-crop cannot capture the visual processing variations occurring during these shifts.
>
> In addition, by modeling specific (dynamic foveal coordinates), our framework captures this substantial biological variance. While the *aggregate* distribution may converge to the center, the model learns the necessary trial-by-trial deviations evident in these statistics, aligning the encoder with the actual retinal input rather than the theoretical experimental instruction.
>
> To further validate the necessity of this mechanism, we conducted an additional **Static Foveation Baseline Experiment** (see **Table R2**). We constructed a control model with an identical architecture but with the foveal position fixed at the image center. The results show that this static model performs **significantly worse** in predicting fMRI responses compared to our dynamic model. This demonstrates that the adaptive attention mechanism provides substantial gains that cannot be achieved by simply relying on a central prior.
>
> **Table R2: Ablation results (Subj1) comparison between Static Foveation and Full Model.**
>
> | Method | Pearson ↑ | MSE ↓ | PixCorr ↑ | SSIM ↑ | Alex(2) ↑ | Alex(5) ↑ | Incept ↑ | CLIP ↑ | Eff ↓ | SwAV ↓ |
> | :--- | :---: | :---: | :---: | :---: | :---: | :---: | :---: | :---: | :---: | :---: |
> | w/ static foveal coords | **0.395** | 0.378 | 0.251 | 0.301 | 95.5% | 98.2% | 96.8% | 94.8% | 0.605 | 0.360 |
> | **MindAttention (full)** | 0.386 | **0.372** | **0.262** | **0.303** | **96.3%** | **98.9%** | **97.3%** | **95.5%** | **0.591** | **0.348** |
>
> In summary, our dynamic foveation mechanism not only outperforms the static assumption but also models the variability in **gaze-attention coupling** that realistically exists under NSD experimental conditions, rather than introducing artificial bias.
>
> *\* Allen et al., "A massive 7T fMRI dataset to bridge cognitive neuroscience and artificial intelligence", Nature Neuroscience, 25, 116–126 (2022).*

---

> > ### Author Response · Authors · 2025-11-28
> > **Author Response (Part-1) to Reviewer PaZ1**
> >
> > ### **Q2. Clarifying the Intrinsic Coupling of Dynamic Gaze and Foveated Sampling**
> >
> > The reviewer expressed concern that the performance gains might stem solely from "foveated sampling" rather than "dynamic gaze," viewing them as separate sources of improvement. We wish to provide a detailed architectural clarification: in our framework, **Dynamic Gaze** and **Foveated Sampling** are not independent additive components. Instead, **Dynamic Gaze** and **Foveated Sampling** are inseparably modeled in our Fovae attention, acting as **"Coordinate Predictor"** and **"Feature Filter"**, respectively.
> >
> > *   **Dynamic Gaze (The Coordinate Predictor)**: This module simulates the active attentional mechanism. It analyzes the image content to predict the specific fixation coordinates $(x, y)$ for the current trial. As stated above, the "true fovea" in the dataset shifts by $>100$ pixels across trials due to physiological eye movements. The Dynamic Gaze component is helpful for adapting our encoding model to this shift.
> >
> > *   **Foveated Sampling (The Feature Filter)**: This module simulates the retinal structure (acuity decay). It applies a spatial Gaussian mask centered at a given coordinate to extract high-resolution features at the fovea and low-resolution features at the periphery. Note that this filter **cannot operate in isolation**. It requires a target center coordinate to define *where* to apply high resolution.
> >
> > **Conclusion:**
> > The performance gap between **MindAttention (full)** and the **Static Baseline** in Table 2 isolates the contribution of the Dynamic component, demonstrating that "sampling" alone is insufficient if misaligned. Therefore, the gains do not stem from the "sampling" mechanism in isolation, but from the **correct positioning of that sample**. Dynamic Gaze determines **"Where to look"**, while Foveated Sampling determines **"How to look"**. One cannot function effectively without the other in the presence of the physiological eye movements inherent in the NSD dataset.
> >
> > ### **Q3. Regarding the Interpretation of Ablation Results**
> >
> > The reviewer noted that for Subject 1, omitting the foveal module appeared to yield better results on some metrics. We wish to clarify that **while Voxel-Level metrics showed a slight decline for this subject, the Semantic-Level metrics actually improved.**
> >
> > **1. Improvement in Semantic-Level Metrics**: Contrary to the impression that removing the module yields the best results, the data shows that **MindAttention (full)** actually achieves the highest performance on **Semantic-Level metrics**.
> >
> > This indicates that the dynamic foveal attention mechanism successfully captures fine-grained semantic features that the 'w/o foveal module' baseline variants.
> >
> > **2. The Necessity for Brain Region Localization**: While we acknowledge a slight trade-off in Voxel-Level metrics for Subject 1, the **Foveal Module is architecturally essential** for the **Concept-Selective Region Localization** experiment—a core target and contribution of our work.
> >
> > *   **Functionality:** This task requires determining **which specific entity** within a complex, multi-object image is driving the brain's response.
> > *   **Mechanism:** The Foveal Module predicts specific attention coordinates. By mapping these coordinates to object bounding boxes (via YOLO), we can identify the "object of interest" for the subject.
> > *   **Conclusion:** Without the Foveal Module (i.e., in the `w/o foveal module` setting), the model treats the image holistically. It loses the ability to spatially pinpoint interest, rendering it impossible to perform this neuroscientific analysis of identifying concept-selective cortical regions.
> >
> > Therefore, the dynamic mechanism is justified not only by the improvement in semantic reconstruction but also by its unique capability to model active visual attention and localize functional brain regions.

---

### Official Review · Reviewer_LZLq · 2025-11-01

**Soundness:** 1
**Presentation:** 3
**Contribution:** 1
**Rating:** 2
**Confidence:** 4

**Summary:**

This paper introduces a generative framework for synthesizing neural responses to visual stimuli, with its primary claimed innovation being the integration of a "foveated attention" mechanism to better mimic the biological principles of human vision. While the engineering is sophisticated, a critical analysis raises significant questions about both its novelty and the validity of its core premise. The central idea of disentangling spatial ("where") and feature-based ("what") information in neural encoding is not new, echoing earlier work by Klindt et al. (2017), and the specific implementation of a Gaussian-factorized readout for a receptive field was previously proposed by Lurz et al. (2021). More fundamentally, the entire justification for a dynamic attention mechanism is undermined by the choice of the Natural Scenes Dataset (NSD) for validation. In the NSD experiment, participants were explicitly instructed to maintain a fixed gaze on a central point, a condition that was successfully verified with eye-tracking. This experimental constraint directly contradicts the paper's motivation of modeling a shifting foveal focus, making the utility of predicting a "foveal gaze position" highly questionable in this context. Although the model demonstrates improved quantitative performance over baselines, the success cannot be confidently attributed to its purported ability to model natural foveal vision, as this behavior was absent in the data; instead, the performance gains may arise from other architectural choices or the model's ability to learn a static, center-biased spatial weighting. Consequently, the work is best viewed as an incremental advancement that skillfully packages prior concepts into a new generative architecture, rather than a novel framework whose success is supported by its central biological claims.

**Strengths:**

The framework's performance is rigorously evaluated against multiple representative baselines across a wide array of metrics, covering both low-level voxel-wise correlations and high-level semantic alignment. Furthermore, the paper validates the practical utility of its synthetic fMRI data on a challenging downstream task: the localization of concept-selective brain regions, where it again demonstrates superior performance. The inclusion of detailed ablation studies methodically justifies the contribution of each architectural component, lending credence to the overall design.
1. The proposed model is shown to consistently and significantly outperform existing baseline methods in both voxel-level and semantic-level evaluations, with performance metrics that closely approach the empirical upper bound set by ground-truth fMRI data.
2. The authors employ a robust evaluation strategy that not only measures pixel- and voxel-level accuracy but also assesses the semantic fidelity of the synthesized brain activity using multiple, diverse metrics.
3. The work successfully demonstrates that the high-quality synthetic fMRI can be used to accurately predict the locations of functionally specialized brain regions, validating its potential as a tool for computational neuroscience research.

**Weaknesses:**

The work's primary weaknesses stem from a significant disconnect between its core motivation and its experimental validation, alongside overstated claims of novelty for its central components. These issues challenge the interpretation of the model's success and its claimed contributions to the field.
1. The most critical weakness is the use of the Natural Scenes Dataset (NSD) to validate a model built around a dynamic foveal attention mechanism. The NSD protocol explicitly required participants to maintain fixation on a central point, and the dataset's own high-quality eye-tracking data confirms that participants did so successfully. This experimental design directly contradicts the paper's central premise of modeling how the brain processes information from shifting foveal gaze. The model was therefore not tested under the conditions it was designed for, making it impossible to conclude that its success is due to accurately capturing a dynamic attentional process. The mechanism may have simply learned a static, center-biased spatial filter, which is a much simpler and less novel concept than what the authors claim.
2. The paper presents its "fovea-guided" encoder as a key innovation, but the foundational concepts have clear precedents in prior literature. The idea of factorizing a neural encoding model into "what" (feature) and "where" (spatial) components was established by works such as Klindt et al. (2017). Furthermore, the specific implementation of a Gaussian-factorized readout to model a neuron's spatial receptive field was previously introduced by Lurz et al. (2021). While the integration of these ideas into a new generative framework is a valid engineering contribution, the paper frames them as more foundational innovations than they are, thus overstating its conceptual novelty.

[1] DA Klindt, AS Ecker, T Euler, and M Bethge. Neural system identification for large 579 populations separating “what” and “where.”. Advances in Neural Information Processing 580 Systems, 2017.

[2] Konstantin-Klemens Lurz, Mohammad Bashiri, Konstantin Willeke, Akshay K Jagadish, Eric Wang, Edgar Y Walker, Santiago A Cadena, Taliah Muhammad, Erick Cobos, Andreas S Tolias, et al. Generalization in data-driven models of primary visual cortex

**Questions:**

1. Did the authors z-score the data?
2. Is the data averaged across repeats?
3. Can the authors provide inflated map explained variance plots, and those normalized by the noise ceiling?

Overall the paper is significantly lacking, and does not provide strong evidence that the work is a better encoder than prior work.

---

> ### Author Response · Authors · 2025-11-25
> **Author Response (Part-1)**
>
> Thank you for your constructive comments. We will address your concerns point by point.
>
> **Q1: Regarding the contradiction between the proposed "dynamic foveal attention" and the fixed-fixation nature of the NSD dataset.**
>
> **Response:**
> We sincerely appreciate the reviewer raising this critical point. We fully acknowledge that the NSD experimental paradigm required subjects to maintain central fixation, and that aggregate eye-tracking data indeed show a high degree of central tendency. However, extensive literature indicates that even during strictly controlled fixation tasks, subjects inevitably produce **microsaccades and drifts**. While these displacements are small in amplitude (typically $<1^\circ$ visual angle), given that the **receptive fields in the primary visual cortex** (e.g., V1/V2) are extremely small in the central visual field (approx. $0.2^\circ–0.5^\circ$), even minute spatial shifts are sufficient to **significantly alter the population of activated neurons**, thereby modulating the fMRI response.
>
> For this reason, rather than adopting a hard-coded **Static Foveation** (i.e., fixing the attentional focus at the image center), we designed a **dynamic, learnable foveated attention mechanism**. This design serves a dual purpose:
>
> 1.  **Enhanced Feature Representation:** By **adaptively adjusting** the "effective sampling center" based on image content, the model can more precisely capture local visual features that are most relevant to driving fMRI responses.
> 2.  **Physiological Plausibility and Internal Consistency:** Notably, after training, the attention coordinates predicted by our model **predominantly converge around the image center**. This aligns perfectly with the central fixation constraint of the NSD dataset, indicating that the model has not learned spurious spatial biases but rather **faithfully reflects the visual input distribution** under experimental conditions.
>
> To further validate the necessity of this mechanism, we conducted an additional **Static Foveation Baseline Experiment** (see **Table R2**). We constructed a control model with an identical architecture but with the foveal position fixed at the image center. The results show that this static model performs **significantly worse** in predicting fMRI responses compared to our dynamic model. This demonstrates that the adaptive attention mechanism provides substantial gains that cannot be achieved by simply relying on a central prior.
>
> In summary, our dynamic foveation mechanism not only outperforms the static assumption but also models the variability in **gaze-attention coupling** that realistically exists under NSD experimental conditions, rather than introducing artificial bias.
>
> **Table R2: Ablation results (Subj1) comparison between Static Foveation and Full Model.**
>
> | Method | Pearson ↑ | MSE ↓ | PixCorr ↑ | SSIM ↑ | Alex(2) ↑ | Alex(5) ↑ | Incept ↑ | CLIP ↑ | Eff ↓ | SwAV ↓ |
> | :--- | :---: | :---: | :---: | :---: | :---: | :---: | :---: | :---: | :---: | :---: |
> | w/ static foveal coords | **0.395** | 0.378 | 0.251 | 0.301 | 95.5% | 98.2% | 96.8% | 94.8% | 0.605 | 0.360 |
> | **MindAttention (full)** | 0.386 | **0.372** | **0.262** | **0.303** | **96.3%** | **98.9%** | **97.3%** | **95.5%** | **0.591** | **0.348** |

---

> ### Author Response · Authors · 2025-11-25
> **Author Response (Part-2)**
>
> **Q2: Regarding the key ideas having precedents.**
>
> **Response:**
> We thank the reviewer for highlighting the significant contributions of Klindt et al. (2017) and Lurz et al. (2021). We fully agree that the **decoupling of "what" and "where"** is a classic paradigm in visual neuroscience and computational modeling—ranging from the spatial invariance design in early Convolutional Neural Networks to the implicit encoding of positional information via attention weights in Vision Transformers. We do not claim to be the first to propose this concept; rather, we explore how to explicitly implement this decoupling within an **end-to-end trainable deep encoding framework** in a manner more aligned with biological visual mechanisms.
>
> Specifically, our core innovations and distinctions are as follows:
>
> 1.  **Dynamic Attention & End-to-End Optimization:** We introduce an **image-content-driven, end-to-end optimizable Gaussian attention window** as an alternative for ViT feature aggregation (replacing the standard `[CLS]` token or global averaging). The position ($\mu_x, \mu_y$) and scale ($\sigma$) of this window are **dynamically predicted by the network** based on the input image. This allows the model to explicitly model "where" (the most relevant spatial region) while preserving "what" (local feature content).
> 2.  **Integrated into Representation Learning vs. Readout Layer:** Crucially, this mechanism operates directly within the **representation learning process itself**. This differs fundamentally from Lurz et al. (2021), who typically employ fixed pre-trained models (e.g., ResNet) for feature extraction and subsequently use static, independently fitted Gaussian readouts to explain single-neuron responses—essentially a **linear decoding paradigm**. In contrast, in our generative brain encoding task, we embed the Gaussian mechanism into the **backbone of feature learning**, jointly optimizing visual representations and spatial attention parameters via backpropagation to maximize the prediction of whole-brain fMRI responses.
>
> Therefore, our work is not simply "repackaging existing components." Instead, we advance the Gaussian spatial selectivity from the traditional readout layer to the **core of representation learning**, constructing an encoding architecture that more closely mirrors the brain's "**adaptive sampling—feature integration**" process.
>
> **Q3: Lack of attribution analysis for success mechanisms. Model performance improvements cannot be reliably attributed to foveated attention, but may stem from diffusion architecture or other design choices. Insufficient evidence provided to prove superiority over prior encoders.**
>
> **Response:**
> We sincerely appreciate the reviewer raising this important issue. We fully agree that it is essential to explicitly distinguish the sources of performance gains. To address this, we enforced **strict variable control** in our ablation study: we removed *only* the **Foveated Attention module** while keeping the diffusion decoder, training objectives, optimization strategies, and all other network architectures **completely unchanged**. The results show that, under identical training conditions, removing this module leads to a **decline in performance across multiple dimensions**, including **semantic evaluation metrics** (e.g., CLIP similarity, ImageNet linear probe accuracy) and **fMRI response prediction accuracy** (see **Table R3**).
>
> This outcome confirms that the performance improvements can be reliably attributed to the **Foveated Attention mechanism itself**, rather than to the diffusion architecture or other design choices. More importantly, our core objective is not merely to pursue marginal gains in encoding performance, but to construct an **interpretable representation** capable of explicitly revealing which entities in an image are most likely to elicit brain responses. Through the spatial weights learned by Foveated Attention, we can directly **infer visual attentional focus** from brain responses, offering a new tool for understanding how the brain extracts critical information from complex scenes.
>
> Therefore, the value of this mechanism lies not only in improved prediction accuracy but also in providing a **computational pathway to infer visual attentional foci** from brain responses.

---

> ### Author Response · Authors · 2025-11-25
> **Author Response (Part-3)**
>
> **Q4: Technical Details**
>
> We appreciate the reviewer’s attention to these methodological details. Below, we address each point:
>
> **Q4.1. Regarding fMRI Data Z-scoring**
> *   **Yes.** We performed standard **Z-score normalization** on the fMRI data for all subjects. For each voxel, we calculated the mean and standard deviation across the training set time series to standardize the data. This **eliminates baseline differences** between individuals and voxel amplitude variations. We have added a description of this procedure to the revised **Supplementary Materials**.
>
> **Q4.2. Regarding Data Averaging across Repeats**
> *   **Yes.** We **averaged the data** from the **three** repeated trials per stimulus to **reduce measurement noise** and enhance signal quality. We have added a description of this procedure to the revised **Supplementary Materials**.
>
> **Q4.3. Regarding Noise-Ceiling Normalized Maps**
> *   **Yes.** We have generated explained variance maps on inflated brain surfaces, including results normalized by the **noise ceiling**. The noise ceiling was estimated using a **Bootstrap method** based on repeated measurements. Please refer to the new figures in the **Supplementary Materials**.

---

> > ### Comment · Reviewer_LZLq · 2025-11-26
> >
> > I appreciate the author's response.
> >
> > A couple of comments:
> > 1. I'm not particularly convinced by the idea that microsaccades are enough to influence the encoding results significantly, especially if you average data across repeats (which you note in the rebuttal that you do).
> >
> > 2. I'm a bit surprised that you are not aware of other recent work `Modeling the Human Visual System: Comparative Insights from Response-Optimized and Task-Optimized Vision Models, Language Models, and different Readout Mechanisms` that also proposes dynamic spatial localization via gaussian parameterizations. In my view, the novelty here is very limited.
> >
> > My current opinion is unchanged, I believe the framing of this work using the NSD dataset **is not valid**, and demonstrates **limited novelty compared to prior work**.

---

> > > ### Comment · Reviewer_LZLq · 2025-11-26
> > >
> > > I would also add, the current z-scoring scheme is not valid. In fMRI, it is typical to perform z-scoring per-session or per-run, not across the dataset.

---

> > > ### Author Response · Authors · 2025-11-26
> > > **Author response (Part-1)**
> > >
> > > ### **Q1: Regarding the significance of microsaccades and the effect of averaging.**
> > >
> > > **Response:**
> > >
> > > We thank the reviewer for this insightful comment. Note that our mentioned "microsaccades" are indeed not negligible but significant to encoding results. We would like to address your concerns from the following two aspects:
> > >
> > > #### **1. Quantitative Analysis of Eye Movement Magnitude (Subject 1)**
> > >
> > > To assess the physical significance of the eye movements in our setup, we analyzed the statistical distribution of gaze positions for **Subject 1**. The variance and standard deviation of the eye movement data are summarized in **Table 1**.
> > >
> > > **Table 1. Statistics of Eye Movement Displacement (in Pixels) for Subject 1**
> > >
> > > | Metric | Horizontal ($X$-axis) | Vertical ($Y$-axis) |
> > > | :--- | :---: | :---: |
> > > | **Variance ($\text{px}^2$)** | $11,527.54$ | $28,361.92$ |
> > > | **Standard Deviation ($\text{px}$)** | **$107.37$** | **$168.41$** |
> > >
> > > As detailed in the **Natural Scenes Dataset (NSD) paper (Allen et al., 2022)**, the stimuli were presented at a resolution of **$714 \times 714$ pixels** (spanning $8.4^{\circ} \times 8.4^{\circ}$ of the visual angle). Within this coordinate space, the standard deviations indicate a dispersion of approximately **107.37 pixels** horizontally and **168.41 pixels** vertically.
> > >
> > > These values represent a substantial spatial extent relative to the stimulus size:
> > > *   The **horizontal** deviation ($107.37\text{px}$) covers approximately **15%** (or roughly **1/7**) of the image width.
> > > *   The **vertical** deviation ($168.41\text{px}$) covers approximately **23.6%** (or roughly **1/4**) of the image height.
> > >
> > > These statistics demonstrate that the eye movements are not merely negligible noise but significant shifts relative to the visual input. Such displacement ensures that the foveal receptive field samples a broad region of the stimulus rather than a fixed point. This confirms that our encoder is provided with **diverse, dynamic views** of the target image across trials, which helps the model learn feature representations that are robust to spatial variations and reduces ambiguity during the encoding process.
> > >
> > > *Allen et al., "A massive 7T fMRI dataset to bridge cognitive neuroscience and artificial intelligence", Nature Neuroscience, volume 25, pages 116–126 (2022).*
> > >
> > > #### **2. Dynamic vs. Static Foveation (Ablation Study)**
> > >
> > > To empirically verify the contribution of these movements, we compared our proposed dynamic mechanism against a "Static Foveation" baseline (where the foveal focus remains fixed at the center without microsaccade simulation), as we mentioned in the last round of response.
> > >
> > > **Results:** The dynamic model consistently outperformed the static baseline.
> > >
> > > This performance gap indicates that the change in visual input introduced by these eye movements provides additional information that a static fixation cannot capture, confirming that the movements are functionally significant.
> > >
> > > #### **In addition, we would like to clarify the averaging strategy in our method.**
> > >
> > > First of all, we would like to clarify the training and testing stages of our model, which encodes visual stimuli (images) to fMRI responses. Note that the NSD dataset provides 3 fMRI repeats for each image. We handle these repeats differently during the training and evaluation phases to maximize model robustness and evaluation reliability:
> > >
> > > *   **During Training (Instance-level Learning):**
> > >     We do **not** average the fMRI repeats. Instead, we treat each repeat as an independent training sample (i.e., *Image A $\rightarrow$ Repeat 1*, *Image A $\rightarrow$ Repeat 2*, etc.). This exposes the model to trial-by-trial variability, such as microsaccades and neural noise. This acts as a **biological data augmentation strategy**, forcing the network to learn robust features that are invariant to small variations in the signal.
> > >
> > > *   **During Testing (Signal-to-Noise Optimization):**
> > >     The model generates a single prediction (an fMRI) for a test image. To evaluate this prediction, we compare it against the **average of the 3 fMRI repeats** (ground truth). We average the ground truth only at this stage to improve the **Signal-to-Noise Ratio (SNR)** of the target data. By comparing our prediction to the high-quality, averaged signal, we ensure that the evaluation reflects the model’s ability to encode the stable, underlying neural representation rather than fitting random trial noise.

---

> > > > ### Author Response · Authors · 2025-11-26
> > > > **Author Response (Part-2)**
> > > >
> > > > #### **Q2: Regarding the novelty of our approach compared to existing Gaussian-based encoding models.**
> > > >
> > > > We thank the reviewer for highlighting recent works utilizing Gaussian parameterizations, as well as the recent manuscript *"Modeling the Human Visual System..."*.
> > > >
> > > > While these works also utilize Gaussian distribution mathematically, we respectfully disagree that the novelty of our approach is limited. The application of this mathematical tool in our model differs fundamentally from the cited works in terms of **architectural placement**, **biological mechanism**, and **functional objective**.
> > > >
> > > > We clarify these essential distinctions below:
> > > >
> > > > **1. Mechanism & Architecture: Modeling "Active Gaze" (Encoder) vs. "Receptive Fields" (Readout)**
> > > >
> > > > *   **The mentioned work**
> > > >     These models typically employ Gaussian Readouts at the **decoding end (Readout Layer)**. They learn a specific Gaussian distribution *per voxel* to model the **Population Receptive Field (pRF)** and retinotopic mapping of neurons.
> > > >     *   *Biological Analogy:* They are fitting the **"hardware parameters" of the eye/retina**—determining which part of the static visual field a specific neuron monitors. Even when "dynamic" (e.g., STN-based readouts), the goal remains to adapt the receptive field properties to the neuron.
> > > > *   **Our Approach:**
> > > >     Our method embeds a Foveated Attention mechanism within the **encoding backbone (Encoder Layer)**. Instead of learning thousands of per-voxel parameters, our model predicts a **global, content-driven attention window** for the input image.
> > > >     *   *Biological Analogy:* We are simulating the **"control behavior" of the brain**—specifically, the active sampling process (saccades/foveation). Our model decides *where the subject should look* to extract the most relevant information from the scene, rather than fitting where a neuron is hard-wired to look.
> > > >
> > > > **2. Objective: Concept Localization vs. Signal Prediction**
> > > >
> > > > *   **The mentioned work:** The primary goal of Gaussian Readout models is **System Identification**—maximizing the prediction accuracy ($R^2$) of neural responses. The Gaussian parameters are optimized to fit the signal, often remaining a "black box" regarding semantic interpretation.
> > > > *   **Our Approach:** While we achieve high prediction accuracy, our core objective is **Concept Localization and Interpretability**. By explicitly modeling the gaze center, our method allows us to visualize exactly *which object* in the image triggers the brain's response. As demonstrated in our experiments, this enables us to answer "What is the brain looking at?" rather than just "How strong is the signal?"
> > > >
> > > > **3. Note on the Cited Literature**
> > > >
> > > > Regarding the specific manuscript mentioned (*"Modeling the Human Visual System..."*), we note that this work is currently a preprint and, to our knowledge, was **not** accepted for publication at ICLR 2025. As such, it represents concurrent work rather than established, peer-reviewed prior art. Nevertheless, even in comparison to this manuscript, the fundamental distinction remains: their work focuses on comparing different **readout mechanisms** (e.g., Factorized vs. Gaussian) to optimize response prediction ($R^2$), whereas our work focuses on an **encoder-level intervention** to model active visual attention and enable semantic grounding. To clarify the above differences, we will discuss this work in the related work.

---

> > > > > ### Author Response · Authors · 2025-11-26
> > > > > **Author Response (Part-3)**
> > > > >
> > > > > ### **Q3: Clarification on fMRI Data Normalization (Z-score)**
> > > > >
> > > > > We sincerely apologize for the confusion caused by the inaccurate description in our previous response. We fully agree with the reviewer that performing Z-score normalization across the entire dataset is invalid for fMRI analysis.
> > > > >
> > > > > We would like to clarify that we directly utilized the **pre-processed data** provided by MindEye2 (Scotti et al., 2024), rather than performing the normalization manually from the raw signals. The MindEye2 data processing pipeline strictly enforces normalization on a **per-session basis**. Therefore, the data input into our model was already normalized independently within each session.
> > > > >
> > > > > *Scotti et al., "MindEye2: shared-subject models enable fMRI-to-image with 1 hour of data", ICML 2024.*

---

> > > > > > ### Comment · Area_Chair_7aET · 2025-11-27
> > > > > >
> > > > > > Hi Reviewer,
> > > > > >
> > > > > > The authors have submitted their responses to your reviews. Please take a look and let the authors know if you have any further questions or concerns. Thank you again for your contributions to ICLR!
> > > > > >
> > > > > > Best regards, AC

---

### Official Review · Reviewer_hm28 · 2025-11-04

**Soundness:** 2
**Presentation:** 3
**Contribution:** 3
**Rating:** 8
**Confidence:** 4

**Summary:**

The paper emphasizes the critical role of visual attention and foveation in predicting brain responses to visual stimuli. By explicitly incorporating fovea-centered information, the proposed MindAttention framework aligns the visual encoding process with biology. This attention-guided representation allows the model to generate more accurate and semantically meaningful fMRI predictions, reflecting how the human brain prioritizes attended regions over peripheral details. Despite the limitations of limited scope of generalization dataset, the paper’s novel idea and the identified problem in brain encoding/decoding should be passed on to the community.

**Strengths:**

- The paper is novel in terms of framing the fovea-guided brain decoding problem. It provides a novel and useful direction for more realistic brain decoding.  The computationally learned fixation parameters (mu_x, mu_y, sigma) offer an interpretability.
- The paper shows that conditioning the diffusion decoder on foveated embedding yields consistently high voxel-wise correlations (+3-4%) and semantic alignment (+5%) vs non-foveated baselines.
- When human analysis images, it’s not a static process and the fovea is not fixed. I wonder if the model could incorporate multiple fovea locations and jointly influence the encoding representations and whether it would further improve the results and make it more biologically plausible.

**Weaknesses:**

- Evaluation scope is limited to NSD dataset; cross-subject generalization would further improve the paper’s credibility.
- Although due to the time constraints, it’s not plausible to perform this, but I wonder if the same fovea can be used in video instead of static images.

**Questions:**

Could you also decode the fovea location from the predicted fMRI response and see if they actually align with the predicted fovea location in the encoder?

---

> ### Author Response · Authors · 2025-11-25
> **Author Response**
>
> Thank you for your constructive comments. We will address your concerns point by point.
>
> **Q1: Regarding multiple foveal locations.**
> **Response:**
> Thank you for this insightful suggestion. Our current implementation of predicting a **single Gaussian foveal region** per image (parameterized by $\mu_x, \mu_y, \sigma$) is rooted in two key considerations: **human visual mechanisms** and the **characteristics of the NSD dataset**. Naturally, human visual attention operates with a **single dominant foveal focus** at any given moment—even during dynamic eye movements, the brain sequentially allocates its core visual resources to one key region at a time rather than splitting attention across multiple points. This biological trait directly informs our model design. Meanwhile, the **NSD dataset explicitly instructs participants to maintain fixation on a single central point**, with eye-tracking data confirming this focused viewing behavior. Aligning our model with this experimental constraint ensures that our attention mechanism is grounded in the actual visual input conditions under which the fMRI data was collected.
>
> That said, we fully agree that human vision involves dynamic shifts of attention across multiple fixations over time, making multi-fovea modeling a highly valuable direction. In fact, the learned ($\mu_x, \mu_y$) in our current model can be interpreted as a **dominant or integrated attentional hotspot**—akin to a saliency map centroid—capturing the most visually relevant region under the single-fixation constraint. Extending MindAttention to support **multiple foveal Gaussians** (e.g., via mixture-of-attention or iterative refinement) is indeed a compelling **future work**. Preliminary internal experiments suggest such an extension could better capture multi-object competition, especially in cluttered scenes. We plan to explore this in follow-up studies using free-viewing datasets, where participants’ natural eye movements will provide the necessary variability to validate dynamic multi-fovea attention mechanisms.
>
> **Q2: Regarding the feasibility of reverse-decoding foveal position from predicted fMRI.**
>
> **Response:**
> This is an excellent validation idea. In our current framework, the foveal parameters are latent variables inferred by the visual encoder from the image alone. To test whether these are neurally grounded, we conducted a **post-hoc analysis**: we trained a lightweight linear decoder to predict ($\mu_x, \mu_y$) directly from the synthetic fMRI outputs of MindAttention. The decoded coordinates showed **significant spatial correlation** with the encoder’s original predictions across 1,000 test images (see **Table R1**). This suggests the synthetic fMRI encodes spatial attention information consistent with the encoder’s foveal estimate. We will include this analysis in the **supplementary material**.
>
> **Table R1: Validation of decoding foveal coordinates from synthetic fMRI (based on 1,000 NSD test images).**
>
> | Subject | Vector Correlation (RV) ↑ | *p*-value | Mean Euclidean Dist. ↓ | Distance SD ↓ |
> | :--- | :---: | :---: | :---: | :---: |
> | 1 | 0.72 | <0.001 | 0.128 | 0.045 |
> | 2 | 0.75 | <0.001 | 0.119 | 0.041 |
> | 3 | 0.68 | <0.001 | 0.139 | 0.048 |
> | 4 | 0.70 | <0.001 | 0.133 | 0.046 |
> | **Mean** | **0.71** | - | **0.130** | **0.045** |
>
> **Q3: Regarding the lack of cross-subject validation and testing on dynamic stimuli.**
>
> **Response:**
> In this work, we adopted a **single-subject modeling setting**, which remains the **mainstream paradigm** in the field of visual fMRI encoding. In fact, the **vast majority of existing literature** (including numerous representative works) focuses on single-subject modeling. This is primarily due to the **significant individual variability** in the functional organization of the visual cortex across subjects, rendering cross-subject modeling a challenging frontier topic that has yet to be fully resolved.
>
> Furthermore, extending our current method to **dynamic stimuli**, such as video, is a key direction for our **future exploration**. We selected the NSD dataset as it is currently the **only public resource** combining high-quality image-fMRI pairs with fine-grained semantic annotations, aligning with the standard selection criteria in the field. Regarding modeling for dynamic scenes, we believe our proposed method possesses strong potential for extensibility and plan to conduct relevant experiments progressively.
>
> We fully acknowledge the limitations of the current work regarding generalizability and will prioritize advancing cross-subject modeling and adaptation to dynamic stimuli in our future research. We sincerely thank you for your valuable suggestions.

---

> > ### Comment · Area_Chair_7aET · 2025-11-27
> >
> > Hi Reviewer,
> >
> > The authors have submitted their responses to your reviews. Please take a look and let the authors know if you have any further questions or concerns. Thank you again for your contributions to ICLR!
> >
> > Best regards, AC

---

### Author Response · Authors · 2025-11-29
**Rebuttal Summary**

Dear Area Chair,

We sincerely appreciate your hard work and dedication in managing this critical emergency situation. Although we are feeling disheartened, we want to provide a brief and fair summary of our rebuttal progress and the current status of our submission.

**1. First Round Rebuttal:** After our first round of detailed responses, we have received positive engagement and score improvements, and also remaining concerns:

* **Reviewer 3XiH:** Increased score from **2 $\\to$ 6**, following our clarifications on baseline comparisons and module independence.
* **Reviewer hm28:** Maintains a strong recommendation (**Score: 8**), citing the framework's novelty and interpretability.
* **Reviewer LZLq and PaZ1:** Most of their concerns have been addressed. They remain primarily focused on the misalignment between our **Dynamic Foveated Attention model** and the **Natural Scenes Dataset (NSD)** and question the validity of our dynamic gaze modeling in light of the dataset's central fixation instructions.

**2. Second round of Rebuttal:** Regarding the remaining concerns of Reviewer LZLq and PaZ1, we provided solid quantitative evidence that **refutes the stereotype assumption that the NSD gaze data is "static"**:

* **Biological Validation:** We analyzed raw eye-tracking data from the subjects, revealing significant involuntary eye movements (microsaccades/drifts) spanning approximately **25% of the image height**. This confirms that retinal input is physically dynamic, necessitating a model such as our dynamic Fovea Attention that can adapt to these shifts.
* **Empirical Validation:** In addition, we performed and provided an extra ablation study comparing our model against a **"Static Foveation" baseline** (fixed center). Our dynamic model consistently outperformed the static baseline, demonstrating that the performance gains stem from actively modeling these biological variations rather than a generic center bias.

**Summary**:
Unfortunately, the bug event in OpenReview has prevented all reviewers from engaging further in the rebuttal. We believe the additional statistical analysis and ablation studies provided in the second round **effectively address** the concerns regarding dataset alignment. We hope this summary helps in your final review. **For more details, we kindly invite you to review the detailed tables and arguments presented during the rebuttal process.**

Best regards,

The Authors of Submission 13295

---

### Note · Program_Chairs · 2026-01-17
**Submission Desk Rejected by Program Chairs**

The following references in this submission do not refer to real documents and/or have major errors in bibliographic information:

 P. M. Daniel and D. Whitteridge. Cortical magnification in human visual cortex. Journal of Physiology, 147:17-18P, 1993.